# Contrasting evolution of the Arabian Sea and Pacific Ocean oxygen minimum zones during the Miocene

Anya V. Hess [1,2,6] ✉, Alexandra Auderset [3,4,6] ✉, Yair Rosenthal[1], Daniel M. Sigman [5] & Alfredo Martínez-García [3]

Ocean oxygen minimum zones have expanded since the mid-20th century, yet their future remains uncertain. Previous studies show that the eastern tropical North Pacific was well oxygenated during the warm Miocene Climatic Optimum (17.0–14.8 Ma), suggesting better oxygenation under climatic warming. To explore whether this response was global, we reconstruct Miocene oxygenation in the second largest oxygen minimum zone, the Arabian Sea. Trace elements and nitrogen isotopes in planktonic foraminifera show that the Arabian Sea was also better oxygenated during the Miocene Climatic Optimum than today. However, deoxygenation history and establishment of a true oxygen deficient zone following the Miocene cooling lagged in the Arabian Sea, indicating the important role of regional oceanographic processes like proto-monsoon or Tethys outflow. Our study supports future projections of deoxygenation reversals in both oxygen minimum zones, but with more complexity in the Arabian Sea due to competing changes in monsoonal upwelling and influx from marginal seas.

Dissolved oxygen is essential for sustaining marine life and plays an important role in regulating marine biogeochemical cycles. However, over the past 50 years, the oceans have lost >2% of their oxygen as temperature has risen, with the greatest change in areas with already low oxygen concentrations, but deoxygenation patterns are complex and vary by ocean region[1]. These areas of the ocean characterized by low dissolved oxygen levels (e.g., <80 μmol/kg, ref. 2) are typically referred to as oxygen minimum zones (OMZs) or, where oxygen concentrations are extremely low leading to water-column denitrification (i.e., <5 μmol/kg, ref. 3), oxygen deficient zones (ODZs). Variations in the OMZ/ODZ's extents can impact the global carbon and nitrogen cycles (e.g., see refs. 4,5) and drive ecosystem change across trophic levels, including those on which we depend for food[6]. Though the deoxygenation trend is expected to continue until 2100, some areas have begun to re-oxygenate[7] and the tropical north Pacific ODZ has shown strong decadal variations rather than a clear trend[8]. Thus, the long-term response of OMZs/ODZs to climate change remains uncertain[9–11].

The warm Miocene Climatic Optimum (MCO; 17.0–14.8 Ma), with similar temperatures and atmospheric pCO₂ concentrations to those predicted beyond 2100 (ref. 12), can provide insight into OMZ/ODZ response to climate warming on these longer timescales. Previous studies have found that the eastern tropical north Pacific (ETNP) was well oxygenated during the

MCO[13–15], contrary to predictions based on the observed modern deoxygenation trend with increasing temperatures. The increased oxygenation is believed to be due to a reduced latitudinal temperature gradient, resulting in decreased meridional and zonal pressure gradients, and reduced equatorial upwelling, leading to reduced organic matter respiration in the shallow subsurface and ultimately overall higher oxygen concentrations[13,14,16].

Recent deoxygenation trends exhibit notable spatial variability, with disparate responses in different ODZs (e.g., see ref. 17). Consequently, the oxygenation pattern from the Miocene ETNP may not be directly applicable to other major ODZs, such as the Arabian Sea. The primary aim of this study is to test whether the Arabian Sea ODZ responded to Miocene climate warming in a similar manner to the ETNP, and to identify the mechanisms driving any observed differences. To achieve this, we reconstruct temporal and spatial changes in the Miocene Arabian Sea ODZ. We compare Site 730 (ODP Leg 117, 17°43.885'N, 57°41.519'E, 1065.8 m water depth) sensitive to changes in the core of the ODZ, off Oman, with Site 714 (ODP Leg 115, 05°03.6'N, 73°47.2'E, 2038.3 m water depth) more distal to the OMZ, in the Maldives (Fig. 1, Supplementary Fig. 1).

To semi-quantitatively constrain the magnitude of oxygenation, we use three proxies that respond to different oxygen concentrations, including those used previously in the ETNP, which allows for direct comparison. We use

[1]Rutgers, the State University of New Jersey, Department of Earth and Planetary Sciences, Piscataway, NJ, USA. [2]Woods Hole Oceanographic Institution, Department of Geology and Geophysics, Woods Hole, MA, USA. [3]Max Planck Institute for Chemistry, Climate Geochemistry Department, Mainz, Germany. [4]University of Southampton, School of Ocean and Earth Science, Southampton, UK. [5]Princeton University, Department of Marine and Coastal Sciences, Princeton, NJ, USA. [6]These authors contributed equally: Anya V. Hess, Alexandra Auderset. ✉e-mail: dranyahess@gmail.com; a.auderset@soton.ac.uk

**Fig. 1 | Map and cross section showing modern oxygenation and site locations. a** Map of minimum oxygen concentration in the water column of the Indian Ocean, showing the Arabian Sea ODZ. Generated in Ocean Data View v. 5.6.5 (ref. 98) by DIVA gridding 50 × 50 grid cells, signal-to-noise ratio of 50. Arrows indicate surface currents annually (gray) and during the summer monsoon (yellow) and winter monsoon (blue) adapted from ref. 99. Note offshore currents along the western edge of the Arabian Sea during the summer monsoon, responsible for upwelling. **b** Cross section A-A' through the Arabian Sea ODZ and Indian Ocean. Color-filled contours are oxygen concentration in µmol/kg. Sites from this study are large dots and sites with additional data from previous publications are small dots. Dot color matches those used for each site in Figs. 2–4. All oxygen data used is World Ocean Database 2018 (ref. 100). ICW Indian central water, SAMW subantarctic mode water.

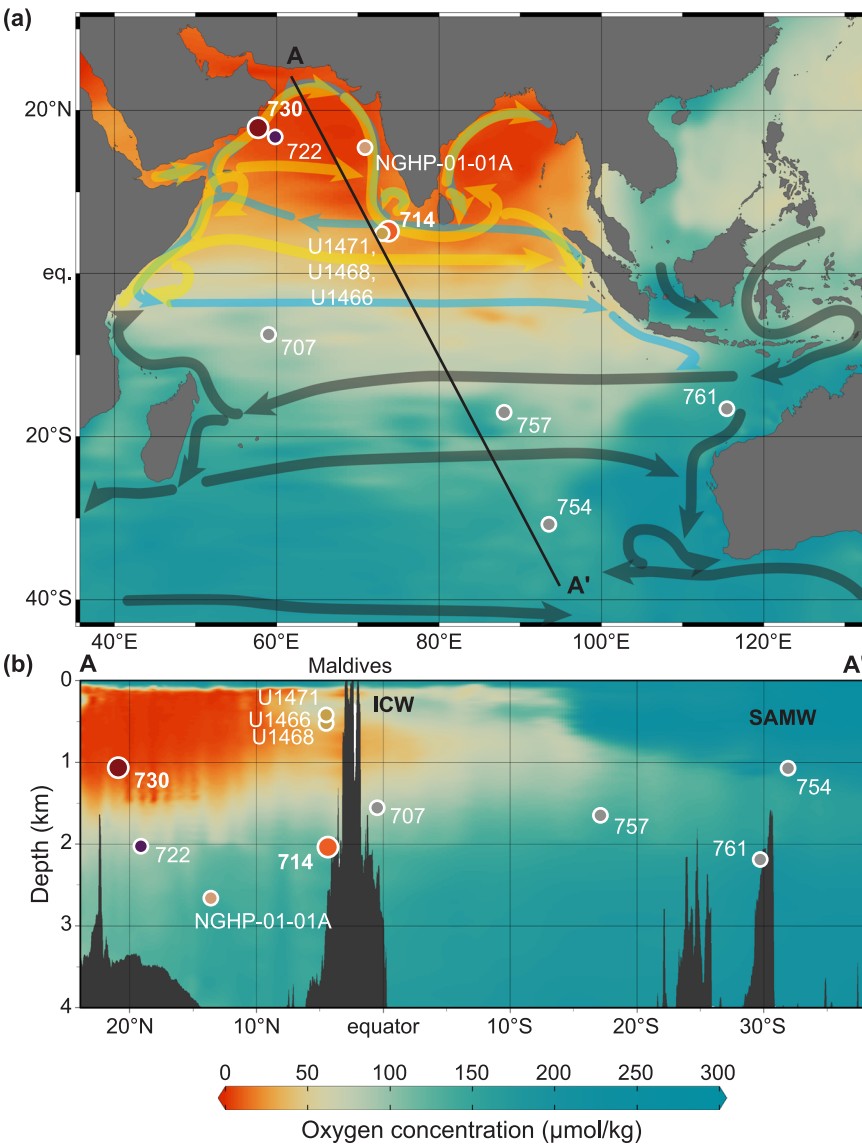

manganese-to-calcium ratios (Mn/Ca) and iodine-to-calcium ratios (I/Ca) in planktonic foraminifera and foraminifera-bound nitrogen isotopes (FB-δ[15]N), which respond to progressively lower oxygen concentrations. Mn/Ca qualitatively tracks bottom-water oxygen in the OMZ range (i.e., <90 µmol/kg)[18], while I/Ca values < 2 µmol/mol exclusively occur in OMZs, where [O₂]$_{min}$ is <90 µmol/kg[19], and FB-δ[15]N responds directly to water column denitrification in ODZs, i.e., oxygen levels <5 µmol/kg[3]. We contextualize deoxygenation trends using Arabian Sea upwelling history derived from sea-surface temperatures (SSTs) and lipid biomarker concentrations, and we compare our findings with data from the Pacific Ocean using the same proxies[13,14] to examine potential drivers of oxygenation in the two largest OMZs.

## Results

A detailed description of how the proxies used in this study function and diagenetic consideration relevant to their interpretation can be found in Supplementary Discussions 1 and 2. Information on age models can be found in Methods and Supplementary Fig. 2.

Mn/Ca ratios in foraminiferal carbonate coatings are indicators of bottom-water oxygenation, with elevated values reflecting oxic conditions that allow for the formation of secondary Mn-carbonate phases in sediments[20]. At Site 714, Mn/Ca values are relatively high between 18.6 and 13.4 Ma compared to before and after: 163.0 ± 87.3 µmol/mol from 19.8 to 18.6 Ma, 212.1 ± 35.2 µmol/mol from 18.6 to 13.4 Ma, and

95.8 ± 20.7 µmol/mol from 13.4 to 8.6 Ma (Fig. 2b, Supplementary Fig. 3). The similarity between Mn/Ca and Fe/Ca signatures (Supplementary Fig. 3b, d) suggests authigenic overgrowth, and we infer that the Mn/Ca signatures are dominated by secondary Mn carbonate in these samples, which were not reductively cleaned (e.g., see refs. 21,22). We therefore interpret the elevated Mn/Ca values from 18.4 to 13.4 Ma as evidence for oxygenated bottom waters coupled with sufficient export productivity to cause reducing conditions in the sediments.

I/Ca ratios in planktonic foraminifera are sensitive indicators of low-oxygen conditions, with values below ~2 µmol/mol corresponding to dissolved oxygen concentrations <90 µmol/kg, based on the species-independent calibration of ref. 19. Throughout the study interval, I/Ca values at both sites are relatively low. At Site 730, I/Ca values are <1.39 µmol/mol throughout the 11.4–15.7 Ma study interval, averaging 0.96 ± 0.21 µmol/mol (Fig. 2c, Supplementary Fig. 3a). At distal Site 714, I/Ca values are slightly higher, <2.06 µmol/mol throughout the 19.8–8.6 Ma study interval, averaging 1.35 ± 0.35 µmol/mol (Fig. 2c, Supplementary Fig. 3a). The offset between sites may be due to the different species used at each site; *Dentoglobigerina altispira* measured at Site 730 calcifies in the deep mixed layer[23], closer to the OMZ and therefore possibly prone to record lower I/Ca values than *Trilobatus sacculifer*[19], which were measured at Site 730. I/Ca values from shallow-subsurface species *Dentoglobigerina venezuelana* from Site 714 are similar to those from *T. sacculifer* (Supplementary

**Fig. 2 | Oxygenation proxies showing timing of deoxygenation across the basin following the MCO. a** Oxygen isotopes from benthic foraminifera[101], for climatic context. **b** Mn/Ca measured on bulk sediment in Site 722[36] and Sites U1466, U1468, U1471[37] and measured on *T. sacculifer* in Site 730 (this study) (**c**) I/Ca measured on *T. sacculifer* in Site 714 and *D. altispira* in Site 730, and (**d**) FB-δ[15]N measured on *T. sacculifer* in Sites 714 and 730. As indicated by the numbers in each sub-panel, each proxy shifts first in the core of the ODZ then more distally, and that shifts occur first in Mn/Ca, followed by I/Ca, then δ[15]N. Note Site 730 seems to have two onsets that indicate denitrification and that Site 714 does not represent local water-column denitrification, instead recording global pycnocline δ[15]N.

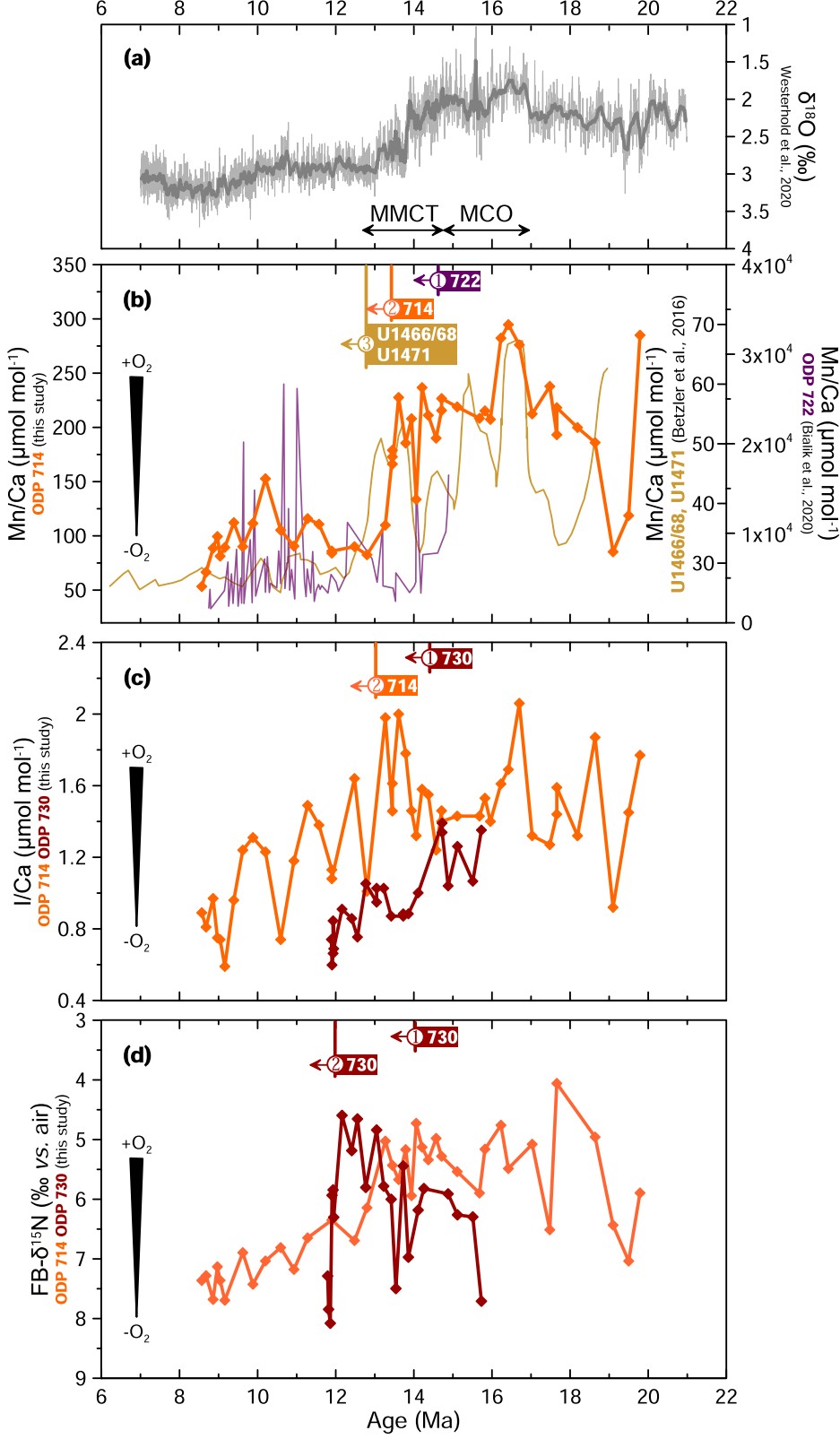

Fig. 3a), which is expected given the overall low I/Ca values[19]. With different mixed layer species at the two sites, we caution against overinterpreting differences in absolute I/Ca values between sites and instead focus on trends through time. Following the MCO, I/Ca values at both sites decrease by ~0.5–1 μmol/mol, beginning first at ~14.3 Ma at Site 730, then at ~13.0 Ma at Site 714 (Fig. 2c, Supplementary Fig. 3a). I/Ca has been shown to record

relatively small oxygenation shifts on the order of 10s of μmol/kg[19], and we interpret the post-MCO I/Ca decreases as evidence for progressive deoxygenation on that order.

Nitrogen isotopes can be used as a proxy for water column denitrification, which occurs under extremely low oxygen conditions (< 5 μmol/kg). Elevated δ[15]N values indicate intensified denitrification and expansion

of the oxygen-deficient zone (ODZ). At Site 730, Miocene FB-$\delta^{15}$N in *T. sacculifer* is 4.6–9.9‰. Two relatively high values >7.0 ‰ between 14.0 and 13.5 Ma are followed by low values (~5.3 ± 0.5 ‰) until ~12.1 Ma when they rapidly increase to >7 ‰ (Fig. 2d, Supplementary Fig. 4a). The high *T. sacculifer* values at ~14.0–13.5 Ma are supported by a value of 9.4 ‰ in *P. mayeri* at 13.9 Ma (Supplementary Fig. 4a). *T. sacculifer* N content averages 3.7 ± 0.4 nmol N/mg calcite (Supplementary Fig. 4b). At Site 714, Miocene FB-$\delta^{15}$N in *T. sacculifer* vary between 4.1 and 9.2 ‰, with the lowest value at 17.7 Ma and a gradual increase starting at 12.8 Ma until 8 Ma, reaching highest values at ~0.2 Ma (Fig. 2d, Supplementary Fig. 4c). N content averages 3.5 ± 0.5 nmol N/mg calcite for *T. sacculifer* (Supplementary Fig. 4d). To minimize potential species effects, we focus on *T. sacculifer* in the main figures, though other species and bulk sediment measurements show consistent trends (Supplementary Fig. 4).

Glycerol dialkyl glycerol tetraether (GDGT) lipids, particularly isoprenoid GDGTs, are commonly used to reconstruct past ocean temperatures, but can also be used to infer changes in marine productivity and preservation conditions[24]. Elevated GDGT concentrations may reflect increased export production or improved preservation under reduced bottom-water ventilation[25]. At Site 730, GDGT concentrations range between 81145 and 793 ng/g sediment with highest Miocene values at 9.7 Ma and lowest at 15.1 Ma (Supplementary Fig. 5). Isoprenoid GDGTs are ~98% isoprenoid and ~2% branched. Crenachaeol averages ~60 % of the isoprenoid GDGTs. The GDGT concentration increases rapidly at ~12.0 Ma and reaches a plateau at ~10.0 Ma (Supplementary Fig. 5), which may be related to increased productivity or decreased bottom-water ventilation and thus better preservation[24–26]. %GDGT$_{RS}$, an index used to assess the influence of the unique Red Sea-type archaeal population[27,28], is high (33–42%) before 13.5 Ma, indicating a stronger contribution of these Red Sea-type GDGT producers, then begin to decrease, reaching ~20% by the end of our Miocene record at 9.8 Ma (Fig. 3c, Supplementary Fig. 5).

## Discussion
### Arabian Sea hypoxia during the Early Miocene and MCO
The combination of I/Ca and FB-$\delta^{15}$N records indicate that an OMZ persisted in the Arabian Sea through the Early Miocene and MCO. I/Ca values generally <2 µmol/mol at ODP Sites 730 and 714 throughout the 19.8–8.6 Ma study interval indicate hypoxic conditions with minimum oxygen concentrations <90 µmol/kg[19]. However, low FB-$\delta^{15}$N values suggest that, unlike today, oxygen levels were not low enough (i.e., <5 µmol/kg) to sustain water-column denitrification in the Arabian Sea. Thus, Arabian Sea oxygen levels were <90 µmol/kg but >5 µmol/kg, so the region was hypoxic but not suboxic as it is today.

Hypoxic conditions in the Early Miocene are broadly consistent with reports of proto-monsoonal activity based on increased terrestrial input in the open ocean Arabian Sea since the latest Oligocene[29]. However, direct records of coastal upwelling during that time are lacking. Alternatively, Early Miocene Arabian Sea hypoxia could be related to low-oxygen Tethys Ocean outflow. Beginning at ~19.5 Ma, a warm, saline water mass originating in what would become the Mediterranean Sea entered the Arabian Sea via the Red Sea and Persian Gulf[30]. Benthic foraminifera assemblages from both the Indian and Atlantic Oceans suggest a shift to low-oxygen conditions beginning ~19.5 Ma, also implying that Tethys outflow waters were low in oxygen[31]. Organic-rich sapropel-like layers in a section from Malta spanning 19.2–18.6 Ma likewise suggest that the Tethys may have been anoxic at that time[32], though these deposits do not appear until later (~15–16 Ma) in the eastern Mediterranean[33,34] and low sedimentation suggests low productivity in that area[35].

### Deoxygenation following the MCO
Further deoxygenation between ~14.6 and 12.1 Ma, during and after the Middle Miocene Climate Transition (MMCT) that followed the MCO, is indicated by multiple proxies across the Arabian Sea (shown by proxy in Fig. 2 and by site in Supplementary Fig. 6). The Mn/Ca, I/Ca, and FB-$\delta^{15}$N proxies respond to different levels of oxygenation due to differences in the

reduction potential of iodine and manganese and the biologically driven threshold for denitrification (Supplementary Discussion 3 for details). Deoxygenation following the MCO was first recorded in coastal sites in what is today the core of the modern OMZ (Sites 730, 722) and, for each proxy, ~1.2 Myr later at distal sites in the Maldives (Sites 714, U1466, U1468, U1471), ~2200 km away (Fig. 2). Overall, the growth of the OMZ, from the first indication of deoxygenation from Mn/Ca in the core of the OMZ to the final rise in nitrogen isotopes, encompassed 2.5 Myr (Fig. 2).

This can be seen as a shift first in Mn/Ca, with Mn/Ca$_{bulk}$ falling at ~14.6 Ma at proximal Site 722[36], then Mn/Ca$_{foram}$ at ~13.4 Ma at distal Site 714 (this study), then Mn/Ca$_{bulk}$ at ~12.8 Ma at distal sites U1466, U1468, and U1471[37](Fig. 2b). The lag of the Mn/Ca shift between Site 714 and nearby Sites U1466, U1468, and U1471 may be because Site 714 is deeper (2038 m), lower in the OMZ, and the other sites are at ~475 m water depth, within the incursion of ventilated Indian central water[38,39] (Fig. 1b). Open-ocean Sites 757, 754, and 752 record decreases in Mn flux much later at ~6.7 Ma[40]. As oxygen levels continued to decline, iodate conversion to iodide intensified, and I/Ca values fell at ~14.3 Ma at ODZ Site 730 then at ~13.0 Ma at distal Site 714 (Fig. 2c). FB-$\delta^{15}$N values rise briefly at ~14.0 Ma at ODZ Site 730, coinciding with a significant cooling step during the MMCT, but they decrease again before rising sharply at ~12.1 Ma. We interpret the ~14.0 Ma event as an initial failed attempt to establish an ODZ. The dramatic increase at ~12.1 Ma likely marks the development of the Arabian Sea ODZ, with persistent water-column denitrification and oxygen concentrations <5 µmol/kg (Fig. 2d, Supplementary Discussion 3). In contrast, the more gradual FB-$\delta^{15}$N increase beginning ~12.8 Ma at distal Site 714 aligns with the rise in Pacific denitrification (Fig. 4c) that appears as elevated $\delta^{15}$N values in the global pycnocline, as indicated by the synchronous rise in FB-$\delta^{15}$N observed in the Atlantic Site 516 (Supplementary Fig. 7). This suggests that Site 714 reflects changes in mean ocean nitrate rather than local water-column denitrification processes during this time.

### Contrasting responses of Pacific and Arabian Sea OMZs to Miocene climate change
Notably, I/Ca values indicate that both the ETNP[13–15] and Arabian Sea (this study) were better ventilated during the MCO than they are today, meaning that the largest ODZs of the world both contracted during the MCO (Fig. 4). However, major differences exist in the timing and magnitude of their response to Miocene climate change. The ETNP was well oxygenated during the MCO[14]. In contrast, this study shows based on I/Ca that the Arabian Sea remained hypoxic throughout the 19.8–8.6 Ma study interval, with oxygen concentrations <90 µmol/kg during the MCO when the ETNP was well oxygenated. Deoxygenation following the MCO may have occurred at about the same time in the Arabian Sea and ETNP, with the I/Ca shift beginning at 15.1 Ma in the ETNP[13,14] and in the Arabian Sea at 14.3 Ma but with age uncertainty of 15.0 to 13.8 Ma (this study) (Fig. 4b). However, we identify the onset of persistent water-column denitrification in the Arabian Sea at ~12.1 Ma, ~2.6 Ma after the ETNP[13,14] (Fig. 4c).

Overall, major differences exist in the oxygenation history of the two OMZs: the Arabian Sea was generally better-ventilated during the Late Oligocene–Early Miocene than the ETNP, had lower oxygen concentrations during the MCO, and full deoxygenation was delayed following the MCO. Thus, though the pattern of increased oxygenation during the MCO followed by deoxygenation suggests that global climate may have played a role, the differences between the two OMZs suggest that regional processes dominated the Arabian Sea oxygenation history during the Miocene.

### Regional (de)oxygenation mechanisms in the Arabian Sea
The modern Arabian Sea ODZ is sensitive to monsoonal wind-driven upwelling and convective mixing[41–43]. During the Miocene, we show that intensified upwelling following the MCO contributed to the observed pronounced Arabian Sea deoxygenation. To reconstruct Miocene Arabian Sea upwelling, we integrated our new Mg/Ca records from surface-dwelling planktonic foraminifera with existing Mg/Ca and TEX$_{86}$ SST records. Modern monsoonal upwelling along the Somali-Oman coast leads to ~6 °C

**Fig. 3 | Relationship between upwelling, ocean circulation, productivity, and oxygenation in the Arabian Sea. a** Oxygen isotopes from benthic foraminifera[101], for climatic context. **b** Sea surface temperature (SST) for the core of the Arabian Sea in red, including Sites ODP 722[46], ODP 730 (this study, [13,46]), and distal Arabian Sea/open Indian Ocean sites in orange, including Sites NGHP-01-01A[48], ODP 714 (this study), ODP 754[13], and ODP 761[47]. The SST gradient between the two areas is indicated in black. See Supplementary Fig. 8 for more detail on the SST gradient calculation. **c** %GDGT$_{RS}$ (pink) indicating the decreasing influence of Red Sea-type GDGT distributions and increase of εNd (gray) in a terrestrial section in Malta[65] indicating restriction of Tethys outflow waters. **d** GDGT concentration (isoGDGT+brGDGT, red line with red filling), FB-δ$^{15}$N *T. sacculifer* (red diamonds) and FB-δ$^{15}$N *P. mayeri* (pink squares) at ODP Site 730 (this study) showing increases in Arabian Sea productivity and denitrification. **e** εNd (gray and black) for ODP Sites 707 and 757[102] indicating reorganization of Indian Ocean circulation. I/Ca from Site 714 (orange) shown for context. Bars at the top of the figure represent hypoxia (gray) and anoxia/denitrification (black). Note that δ$^{15}$N scale is reversed compared to Fig. 2.

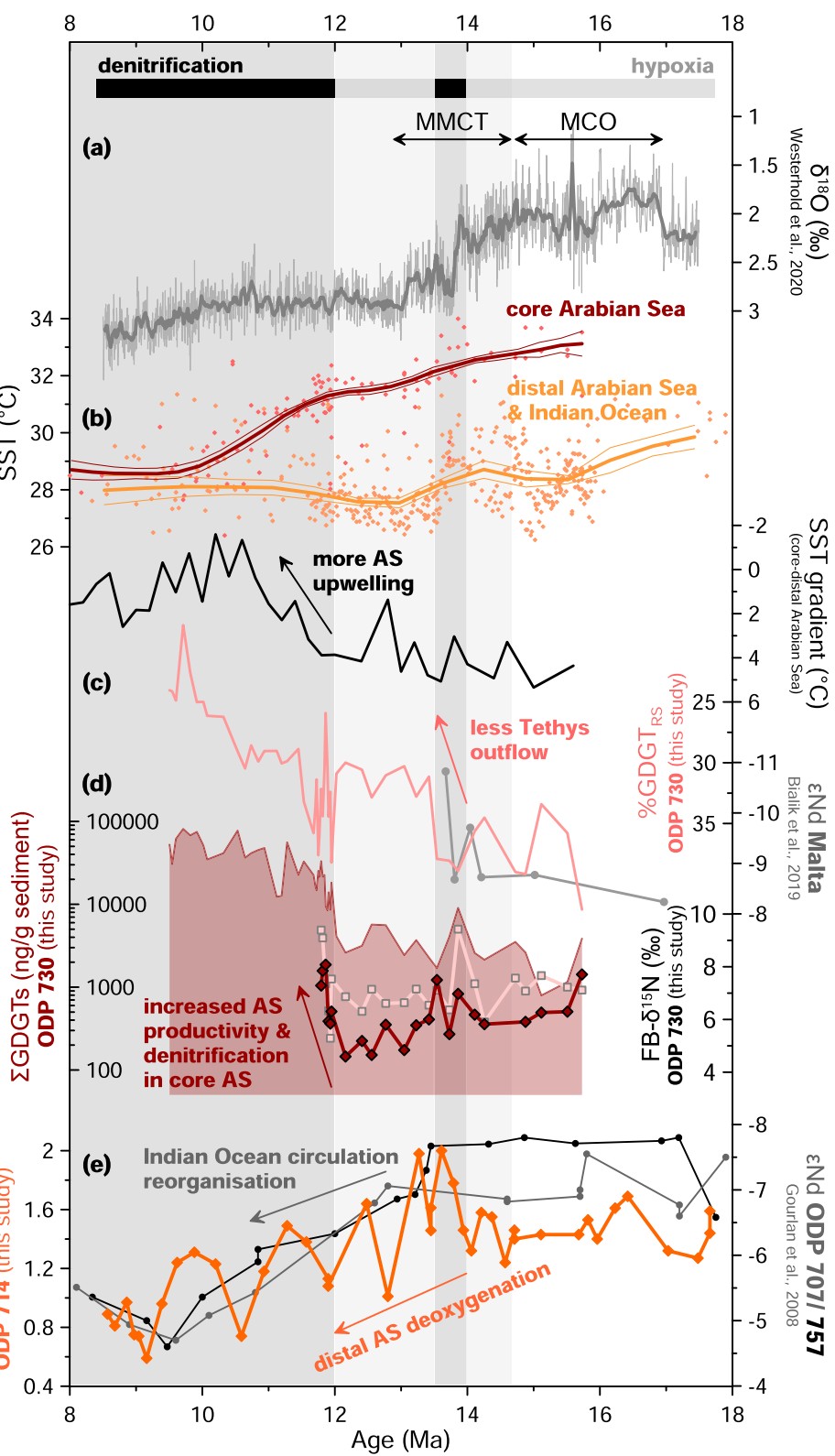

of seasonal SST cooling[41,44,45]. A similar pattern emerges beginning at ~14 Ma, when SSTs in the OMZ cooled relative to those more distal to the OMZ (Fig. 3b). A two-step SST cooling at ~14 and 12 Ma of ~6 °C can be seen in the western Arabian Sea (ODP Site 730 and 722, Mg/Ca from this study and TEX$_{86}$ from refs. [13,46]). In contrast, despite some local heterogeneity, SSTs were generally cooler and less variable in the sites more distal to the OMZ in the eastern Arabian Sea and Indian Ocean (ODP Sites 714, NGHP-01-01A, 754, and 761 from this study,[13,47,48]) (Fig. 3b and

Supplementary Fig. 8). The cooling at ~14 Ma is paralleled in the distal composite from records from Site NGHP-01-01A in the eastern Arabian Sea, where cooling is attributed to increased monsoonal activity[48], and Site 761 in the eastern Indian Ocean, where it is attributed to reduced influx of warm waters from the shallow Indonesian Throughflow, which decreased as Antarctic ice growth lowered sea level[49]. Lipid biomarker (GDGT) concentrations in Site 730 also increase in two steps, indicating enhanced productivity or better biomarker preservation under low-oxygen

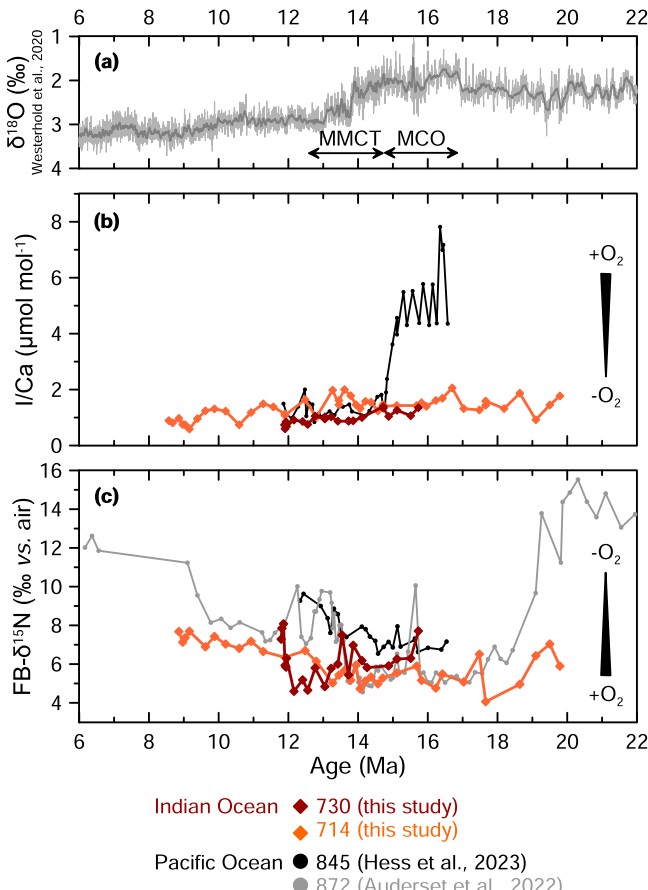

**Fig. 4 | Inter-basin comparison of oxygenation histories for the Arabian Sea and Pacific Ocean. a** Oxygen isotopes from benthic foraminifera[101], for climatic context. **b** I/Ca data from the eastern tropical North Pacific (ODP 845 *D. altispira*[14]) and Arabian Sea (ODP 730 *D. altispira* and ODP 714 *T. sacculifer*, this study). **c** FB-δ[15]N data from symbiont-bearing planktonic foraminifera in the Pacific Ocean (ODP 872[13], ODP 845[14]), and Arabian Sea (ODP 730 and ODP 714, this study). Arabian Sea shown in orange/red and Pacific Ocean in grey/black.

conditions[24-26] (Fig. 3d and Supplementary Fig. 5). This trend is also seen in the total organic carbon content in Site 730, peaking around 12–9 Ma[50]. In addition, upwelling intensification is supported by the appearance of *Globigerina bulloides*, a planktonic foraminifera that thrives in upwelling conditions, at Site 730 at ~12.9 Ma[50], and by upwelling-related zooplankton communities at nearby Site 722 at ~13.4 (ref. 51), as well as a benthic δ[13]C shift to lighter values that indicates the influence of old (upwelled) bottom waters at Site 730[50] (Fig. 5b).

Intensified western Arabian Sea upwelling coincides with reported intensification of monsoonal activity and establishment of the modern South Asian Monsoon system (Fig. 5c)[52]. Model evidence suggests that changes in regional topography, particularly the emergence of land in the Arabian Peninsula region and Iran-Zagros topography, played a key role in reorganizing southwesterly winds of the Somali Jet and upwelling patterns in the Arabian Sea around 13 Ma. The spread of C4 plants in Asia at 15.5–14 Ma has been linked to increased seasonality and summer precipitation[53] (Fig. 5b). At Site NGHP-01-01A in the eastern Arabian Sea, increasing rainfall intensity from ~15 to 11 Ma is inferred from foraminiferal Ba/Ca and oxygen isotopes[48], leading to increasing stratification as indicated by Mg/Ca-derived temperatures from foraminifera living at different water depths[54]. Together, these findings suggest increased upwelling and primary production, likely connected to monsoonal intensification[36,37], which could explain the delayed but progressive post-MCO deoxygenation.

Convective mixing, driven by surface cooling and increased water density, is another mechanism that can influence oxygenation in the upper ocean of the Arabian Sea. Nowadays, this process is most active during winter monsoon seasons, when strong winds enhance evaporative cooling and promote vertical mixing[42,43]. Similar dynamics have been inferred for the Last Glacial Maximum (LGM), during which intensified winter monsoonal winds led to substantial surface cooling and deeper convective overturning, entraining nutrient-rich waters into the surface layer and potentially enhancing productivity[55]. The global cooling and temperature reorganization during the MMCT could have led to intensified convective mixing and nutrient supply to the surface, promoting ODZ growth. However, due to the lack of seasonal proxies in our study interval, this mechanism remains speculative.

In addition to regional upwelling and convective mixing, broader circulation changes may have also played a role in shaping the Arabian Sea ODZ. During MMCT cooling, atmospheric cells in the southern hemisphere shifted northward. In the tropics, a northward shift of the intertropical convergence zone (ITCZ) between 14.6 and 12 Ma[56,57] would have led to strengthening of the southwesterly winds along the western Arabian Sea and intensified monsoons, strengthening the OMZ[36]. The southern hemisphere westerlies also shifted northward and intensified by ~12 Ma[57-59]. This circulation helps drive intermediate waters from the Southern Ocean to ventilate the thermocline and upper intermediate depths globally[60]. If the biological pump in the Southern Ocean strengthened during the MMCT as it did during Pleistocene glacials[61], this and a shift in the westerlies could have altered the fluxes of nutrients and oxygen to the low latitude mid-depths as proposed by ref. 13. This shift in the westerlies coincides with our observed increases in lipid biomarkers and FB-δ[15]N at Site 730 and rising diatom productivity at Site 722[51], suggesting that both regional upwelling and increasingly nutrient-rich, oxygen-poor ocean interior waters contributed to ODZ intensification at ~12.1 Ma.

The dynamic tectonic setting of the Miocene also led to shifting ocean gateways, causing dramatic changes in ocean circulation and likely affected the OMZs on a regional scale. The Arabian Sea is semi-enclosed by the Arabian Peninsula and India, leading to restricted deep-water exchange and oxygen replenishment, whereas the ETNP, in the open Pacific Ocean, has a better deep-water circulation. In theory, this alone could lead to differences in OMZ dynamics and potentially explain the lower oxygen concentrations in the Arabian Sea compared to the ETNP.

Additionally, we find strong evidence for the presence of warm, saline, low-oxygen Tethys outflow waters (Fig. 6a). Prior to ~13.5 Ma, Site 730 exhibits high %GDGT_RS values indicating a high proportion of Red Sea-type archaeal communities (this study, Fig. 3c). SSTs are also consistently warmer in the western Arabian Sea than in more distal sites that are closer to the equator (Fig. 3b). This is supported by previous authors who showed that the Arabian Sea was likely affected by Tethys outflow waters during the Middle Miocene, based on stable isotopes[30,31,62-64] (Fig. 5d). Low-oxygen Tethys outflow may explain why the Arabian Sea remained more hypoxic than the ETNP during the MCO and, in combination with other factors, could explain the false start to denitrification at ~14.0 Ma.

We propose that the complex deoxygenation history following the MCO was the result of competing forcing from increasing upwelling, restriction of low-oxygen Tethys outflow, decreased stratification due to colder surface waters during global cooling, and increasing nutrient supply and decreasing oxygen supply through Southern Ocean-sourced deep waters, themselves driven by both climatic and tectonic forcings. The FB-δ[15]N increase at ~14.0 Ma indicates increased water-column denitrification resulting from the initial strengthening of Arabian Sea upwelling, possibly linked to the global cooling step during the MMCT (Fig. 6b). At ~13.5 Ma, low-oxygen Tethys outflow was constricted by a drop in sea level, as indicated by a decrease in εNd at Malta[65] and supported by a dramatic decrease in our %GDGT_RS (Fig. 3c). This apparently counteracted the effects of increased upwelling, increasing oxygen levels enough to shut down water-column denitrification. The final FB-δ[15]N rise at ~12.1 Ma coincides with a more rapid upwelling increase (Fig. 6c), decreasing oxygenation and

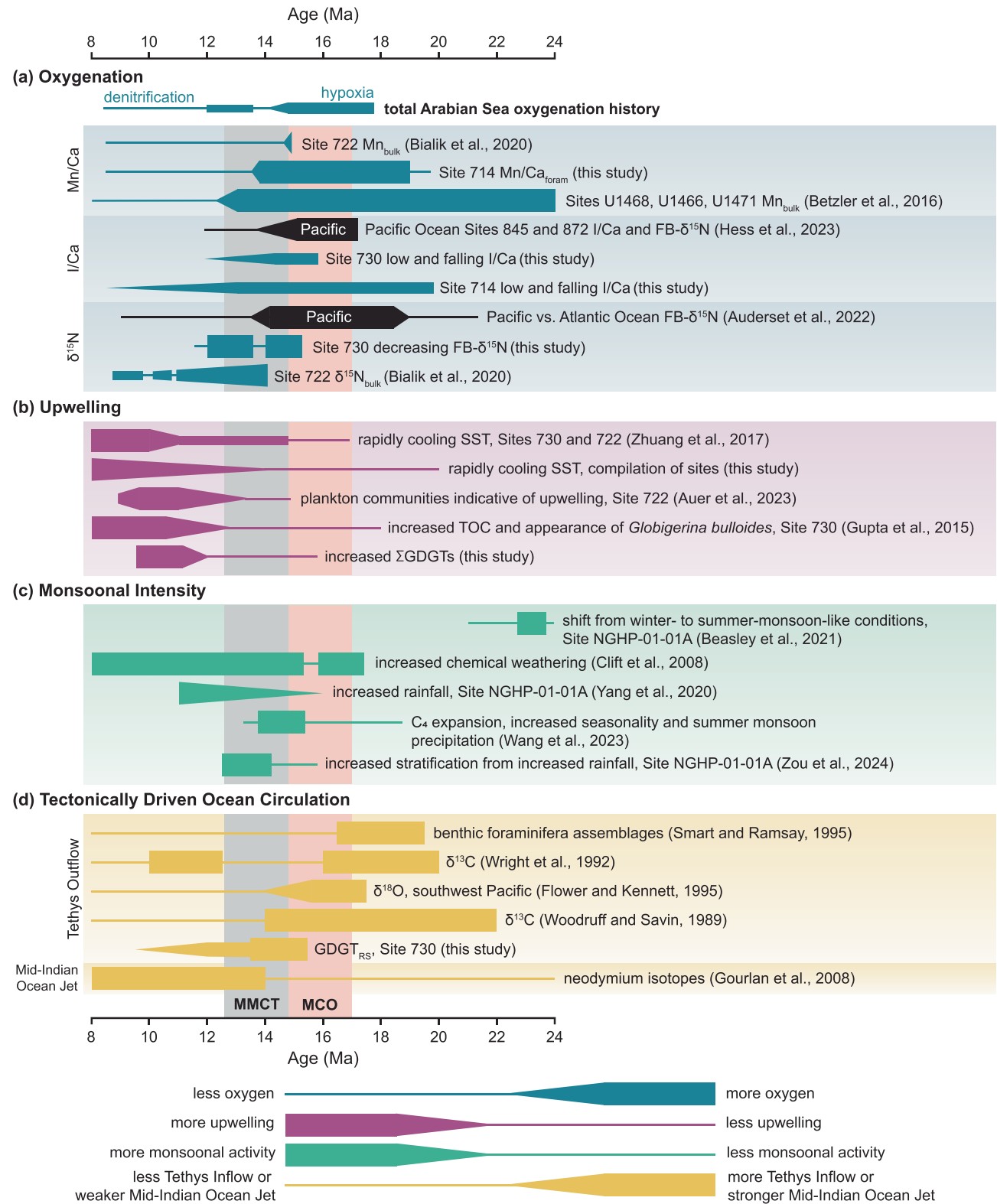

**Fig. 5 | Summary of oxygenation and factors affecting it in the Arabian Sea and Pacific Ocean. a** Oxygenation. **b** Upwelling. **c** Monsoonal intensity. **d** Tectonically driven ocean circulation. Increasing thickness indicates relatively more oxygen or more monsoonal activity, scaled separately for each publication. Timing follows authors' published ages. Red bar indicates Miocene Climatic Optimum (MCO), gray bar indicates Middle Miocene Climate Transition (MMCT). Based on data from this study and refs. 13,14,29–31,36,37,46,48,50,51,53,54,62,102–104.

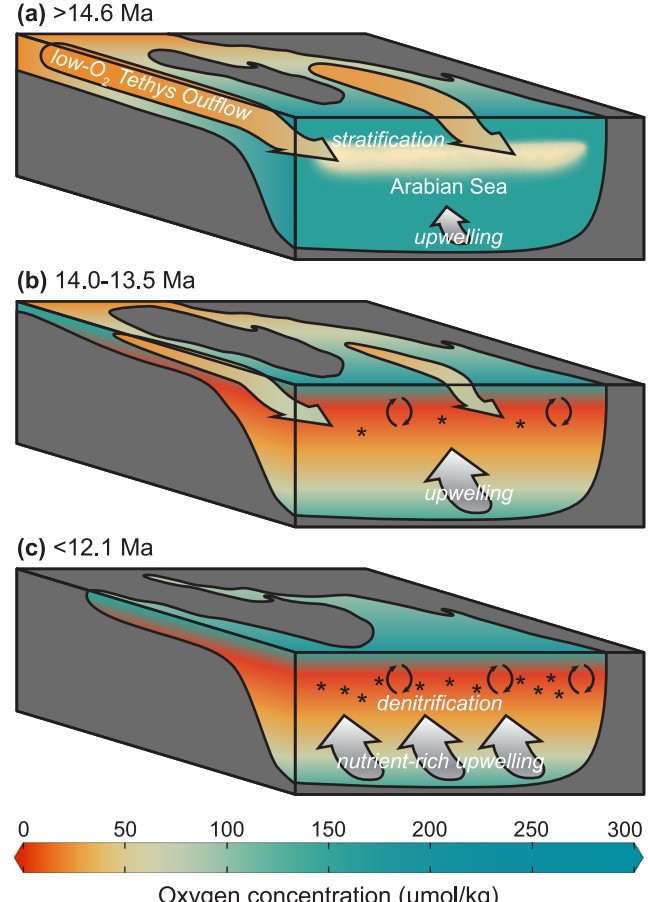

**(a) >14.6 Ma**

low-O₂ Tethys Outflow

stratification

Arabian Sea

upwelling

**(b) 14.0-13.5 Ma**

upwelling

**(c) <12.1 Ma**

denitrification

nutrient-rich upwelling

Oxygen concentration (µmol/kg)

0    50    100    150    200    250    300

**Fig. 6 | Illustration of major timesteps in Miocene Arabian Sea oxygenation history. a** Prior to 14.6 Ma, low-O₂ Tethys outflow and/or proto-monsoonal upwelling leads to [O₂] of 5–90 µmol/kg. **b** 14.0–13.5 Ma, decreasing influx of low-O₂ Tethys outflow is compensated by increasing upwelling and increased convective mixing (circular arrows) due to colder surface temperatures, leading to higher productivity (stars). **c** Beginning at 12.1 Ma, low-O₂ Tethys outflow is decreased or eliminated, and strong upwelling becomes the dominant force, decreasing [O₂] to <5 µmol/kg, leading to persistent denitrification. Increased convective mixing (circular arrows) and increased nutrient input from the Southern Ocean may also have played a role.

leading once again to appreciable water-column denitrification (Fig. 6c). The combination of decreasing stratification and increased nutrient supply leading to increased productivity may have contributed to deoxygenation.

### Links to regional and global climate and implications for the future

Our findings highlight the need to understand regional processes to predict future OMZ change in the Arabian Sea[66]. Climate change is expected to alter water mass exchanges between the Persian Gulf, Red Sea, and Arabian Sea by modifying thermal structure, salinity gradients, and sea level[67]. For instance, future warming may reduce winter mixing in the Persian Gulf, leading to more oxygen-poor outflows[68], while increased buoyancy could limit the ability of these currently oxygen-rich waters to ventilate deeper layers[66]. Earth system models project that stronger land-sea thermal contrasts will enhance summer monsoon winds, potentially boosting upwelling and productivity[69]. However, increased rainfall may intensify surface stratification through freshwater input, suppressing vertical mixing and nutrient supply[70,71]. The Miocene provides a geological perspective that supports models suggesting that continued warming may lead to reduced upwelling due to increased stratification and an important role for low-O₂ outflow waters in Arabian Sea deoxygenation.

Also, the coincident deoxygenation in the Arabian Sea and ETNP following the MCO suggests a role for global climate. This may have been through increased upwelling, accomplished in the Arabian Sea by changes in monsoonal intensity[36,37,48,52–54] and/or a northward shift of the ITCZ[56,57] and in the Pacific by a stronger Pacific Walker Circulation[14]. Another possibility is that a strengthening of the Southern Ocean biological pump decreased the oxygen content of the global ocean interior[13].

## Conclusions

In summary, we present new semi-quantitative geochemical evidence for a long-lived OMZ in the Arabian Sea extending back to at least 19.8 Ma based on persistently low I/Ca in planktonic foraminifera. This suggests that early proto-monsoonal activity and/or the influx of low-oxygen Tethys outflow waters played a role in maintaining oxygen-deficient conditions long before the modern monsoon system developed. Following the MCO, the Arabian Sea experienced deoxygenation, culminating in the establishment of a true ODZ by ~12.1 Ma. This deoxygenation was progressive, with Mn/Ca, I/Ca, and FB-δ¹⁵N proxies each recording successive phases of oxygen loss over a ~2.5 Myr period. A comparison with the ETNP shows that though both areas were better oxygenated during the MCO than today, key differences exist in their Miocene oxygenation histories. Both OMZs deoxygenated following the MCO, but Arabian Sea deoxygenation was more complex, with full deoxygenation and development of an ODZ with denitrification occurring ~2.6 Myr later in the Arabian Sea (~12.1 Ma) than in the ETNP (~14.7 Ma). To the extent that the MCO is an analog for understanding the long-term response of OMZs to climate warming, oxygen concentrations might be expected to rise in both regions to levels that inhibit denitrification. Our findings highlight that the effects of regional oceanographic and tectonic settings may be substantial, an important consideration for predicting the future of ocean oxygenation, particularly in the Arabian Sea where changes in monsoonal upwelling and inflow from marginal seas may be important.

## Materials and Methods
### Study sites and oceanographic setting

Study sites were chosen for (1) temporal coverage of the MCO and the cooling that followed and the Mn depletion and δ¹⁵N_bulk events thought to indicate OMZ expansion[36,37], (2) foraminifera abundance and preservation, and (3) spatial distribution within (proximal to) and more distal to the OMZ.

Regional upwelling close to proximal Site 730 is driven by summer (southwest) and, to a lesser extent, winter (northeast) monsoons. During the summer, southwesterly winds drive upwelling along the Arabian and Somali coasts, and increased runoff further fuels productivity[41]. During the winter, northeasterly winds from the Tibetan Plateau drive upwelling along the southwestern coast of India[72]. Despite the focus of upwelling and productivity in the western Arabian Sea, oxygen limitation occurs throughout the northern Arabian Sea, eastward of the high upwelling and productivity area[73], including towards distal Site 714. This apparent paradox can be explained by introduction of oxygenated waters into the western Arabian Sea from the Red Sea and Persian Gulf or from the south via the Somali Current and eastward transport of organic detritus, resulting in increased remineralization to the east[74,75] (Fig. 1).

Age-depth correlation for Site 730 is based on biostratigraphic occurrence data and strontium isotope chemostratigraphy, and for Site 714 is based on biostratigraphic occurrence data (Supplementary Fig. 2, Supplementary Data 1). Planktonic foraminifera at Site 730 are well to moderately preserved, and at Site 714 are well to poorly preserved, with preservation poorer down-section (dissolution and overgrowth), based on light microscope and scanning electron microscope analyses (Supplementary Fig. 9) and consistent with ODP initial reports[76,77].

### Trace elemental analyses

Trace element analysis was done on 11–20 moderately well-preserved surface-dwelling *Dentoglobigerina altispira* per sample from the

300–355 μm size fraction at Site 730 (Supplementary Fig. 3, Supplementary Fig. 9, Supplementary Data 2). For Site 714, 17–21 surface-dwelling *Trilobatus sacculifer* without a sac from the 300–355 μm size fraction were selected, aiming for the 20-foraminifera per sample suggested for I/Ca analysis[78]. Foraminifera were gently crushed between glass plates to open the chambers. Chemical cleaning and analytical procedures follow the protocol of ref. 79. The protocol consists of the steps usually used for trace elemental analysis: removal of fine particulates using rinses with ultrapure water, removal of clays using rinses with methanol, removal of metal oxides using reductive cleaning with ammonium hydroxide, citric acid/ammonia, and hydrazine (this step was performed on samples from Site 730 but not 714), removal of organic matter using oxidative cleaning with ammonium hydroxide and hydrogen peroxide, and removal of authigenic carbonate using a diluted acid leach. Analysis of samples from Site 714 were performed after ref. 79 was published; they showed that reductive cleaning lowers I/Ca values, and thus that step was not performed on those samples. However, the effects of reductive cleaning are predictable and to correct for these effects on Site 730 samples, Mg/Ca values from that site were multiplied by 1.1 following refs. 22,80 and I/Ca values were multiplied by 1.3 following ref. 79, making values comparable between the two sites.

Foraminifera were dissolved in 0.065 N trace-metal clean $HNO_3$ (OPTIMA) the morning of each analysis run. A 95 μL aliquot of this solution was diluted in 300 μL of trace-metal clean 0.5 N $HNO_3$, resulting in a Ca concentration of 4 ± 2 mmol/L. Samples were analyzed on a Finnigan MAT Element XR Sector Field Inductively Coupled Plasma Mass Spectrometer in low resolution ($m/\Delta m = 300$) using the method of ref. 81. Anhydrous ammonia gas was injected into the high-purity quartz cyclonic spray chamber (Elemental Scientific), raising the pH of injected samples ( >9.14) and reducing the memory effect of iodine and improving washout efficiency. The ammonia gas is also expected to have stabilized iodate in solution[79]. To correct for matrix effects due to varying Ca concentration among samples, six standard solutions with the same elemental ratios but varying Ca concentrations were measured and used to quantitatively correct each sample to the SGS Ca concentration. For numbers reported in the text, plus-or-minus error is 1 standard deviation. Measurement of standard solutions across runs yield RSD%s of 0.3–0.4% for Mg/Ca, 0.6–2.3% for Mn/Ca, 1.8–3.3% for Fe/Ca, and 0.3% for Sr/Ca. Due to the volatile nature of iodine, standard solutions were spiked with potassium iodate the morning of each run; the RSD% for I/Ca is 5–6%, a measure of the standard deviation of the procedure, including preparation and measurement. The black bars in Supplementary Fig. 3 show the standard deviation of replicates from all runs, where replicates are sets of foraminifera from the same sample.

To convert Mg/Ca to temperature, we used Eq. (1):

$$Mg/Ca = 0.38e^{0.09T} \times \left(3.43/5.2\right)^{0.41} \qquad (1)$$

for surface-dwelling foraminifera, in which T is the calcification temperature. This equation is based on the multi-species equation from ref. 82 with a term to correct for Mg/Ca$_{seawater}$ from ref. 83. We used a constant Mg/Ca$_{seawater\ Miocene}$ = 3.43, consistent with recent studies of Mg/Ca during the Miocene[14,49,84]. If we were to use a variable Mg/Ca$_{seawater Miocene}$, increasing through the study interval as in ref. 85, Mg/Ca-derived temperatures would be shifted colder for younger samples by <2 °C, making our estimated cooling conservative. Since this affects both core OMZ sites and non-OMZ sites, the temperature difference between these areas (Fig. 3b SST gradient) would remain relatively unchanged. No correction was made for pH, as Mg/Ca$_{sacculifer}$ has been shown to be insensitive to that parameter[86].

## Nitrogen isotope analyses

For the foraminifera-bound nitrogen analysis (Supplementary Fig. 4, Supplementary Data 3) in Site 730, we picked 80–400 *T. sacculifer*, 90–400 *D. altispira*, and 65–400 *Paragloboratalia mayeri* from the 255–400-μm size fraction. In Site 714, we picked 400 *T. sacculifer*, *Globorotalia menardii*, and *P. mayeri* from the 255–400 μm size fraction. The foraminifera tests were gently crushed with a watch glass in a glass petri dish and cleaned with Na-polyphosphate solution (pH 8) to remove clays, with dithionite-citric acid solution to remove metal oxides, and with potassium persulfate/sodium hydroxide solution to remove external organic matter following the FB-δ¹⁵N protocol by ref. 87, updated by ref. 88.

Next, 2–5 mg of cleaned foraminifera fragments were dissolved in hydrochloric acid and the crystal-bound organic nitrogen was oxidized to nitrate using a basic potassium persulfate solution. 5 nmol nitrogen of nitrate in the sample solution was added to a gas-tight vial with 2.75 mL medium containing a strain of the denitrifying bacteria *P. chlororaphis* that converts the nitrate quantitatively to nitrous oxide[89]. Finally, the δ¹⁵N of nitrous oxide was measured by gas chromatography-isotope ratio mass spectrometry[90–92].

The precision and accuracy of the corrected isotope values was measured using three different in-house (Max Planck Institute for Chemistry) foraminifera and coral laboratory standards: a mixed foraminifera standard (63–315 μm size fraction) from the North Atlantic (MSM58-17-1) with δ¹⁵N of 5.84 ± 0.16‰[93], a coral standard from the taxon *Porites* (PO-1) with δ¹⁵N of 6.2 ± 0.3‰, a coral standard from the taxon *Lophelia* with δ¹⁵N of 10.01 ± 0.4‰[94]. We calibrated the measured δ¹⁵N with international nitrate isotopic references IAEA-NO3 and USGS-34 and corrected for the per-sulfate oxidation blank. The analytical precision for the internal standards across 4 different batches (3 *Lophelia* and 3 mixed foraminifera standard per batch) was ±0.20‰ (*n* = 18) for Lophelia and ±0.24‰ (*n* = 18) for the mixed foraminifera standard. The oxidation blank per oxidized sample was between 0.17 and 0.24 nmol N. The nitrogen content was estimated by using a calibration curve with the international nitrate isotopic references IAEA-NO3 and USGS-34. The analytical precision for the N content in *Lophelia* was ± 0.16 nmol N/mg (*n* = 18) and mixed foraminifera standard ± 0.50 nmolN/mg (*n* = 18). It was not possible to measure replicate samples for foraminifera samples in Sites 714 and 730 due to the low number of for-aminifera. The black bars in Supplementary Fig. 4 show the standard deviation of the δ¹⁵N and N content in mixed foraminifera standards from all 6 batches.

Stable isotope δ¹⁵N$_{bulk}$ of organic samples were analyzed by combustion method, on a Costech elemental analyzer (EA) (ECS4010) interfaced with a ConFlo IV and a Thermo Delta V plus mass spectrometer. Helium was used as a carrier gas, at a flow rate of 100 ml/min. Samples were weighed and encapsulated into tin capsules and stored in desiccators until analysis. The samples were combusted at 980 °C over a Chromium (III) oxide catalyst in the presence of excess oxygen (25 ml/min). Silvered cobaltous/cobaltic oxide was positioned lower in the quartz combustion tube. Any nitrogen oxides were reduced by passage over copper wire (650 °C) to nitrogen gas, and traces of water were removed through a magnesium perchlorate trap. The $N_2$ peak for each sample was separated through a gas chromatography (GC) column (55 °C) before analysis by IRMS. Nitrogen isotopes (δ¹⁵N$_{bulk}$) were calibrated using two-point regression with USGS40 and USGS41, with precision 0.13‰ and 0.10‰, respectively.

## Lipid biomarker analyses

The TEX$_{86}$-derived sea surface temperatures in Site 730 were initially published in ref. 13. However, neither GDGT concentration nor %GDGT$_{RS}$, an index to identify Red Sea-type GDGT distributions, were discussed in the previous publication. In this study, we report the concentration of iso-prenoid GDGTs, branched GDGTs, and crenarchaeol (Supplementary Fig. 5, Supplementary Data 4). Further, we discuss %GDGT$_{RS}$ in context of the geochemical data generated for this study. A detailed description of the lipid biomarker method can be found in ref. 13. In short, we freeze-dried an average of 2.5 g sediment for each of the 55 samples. We extracted and separated the organic compounds into two fractions (non-polar and polar fraction) using the 2-fraction method proposed by ref. 95. Subsequently, we added 60 μl (~540 ng) of an internal standard (C$_{46}$-GDGT, synthesized by ref. 96) for quantification to the polar fraction containing GDGTs. We dried the extracts in a centrifugal Rocket Evaporator (Genevac) and filtered them with a polytetrafluoroethylene filter (0.2-μm pore size) with a 1.4 % mixture

of hexane:isopropanol (hex:IPA). The GDGTs were analyzed with a high-performance liquid chromatographer (Agilent, 1260 Infinity) coupled to a single-quadrupole mass spectrometer detector (Agilent, 6130) generally following the protocol proposed by ref. 97. For the compound separation in the HPLC, we used one ultra-high-performance liquid chromatographer silica column (BEH HILIC column, 2.1 mm × 150 mm, 1.7 μm; Waters) maintained at 30 °C. The flow rate of the 1.4 % hex:IPA mobile phase was 0.2 ml min$^{-1}$ and kept constant for the first 25 min, followed by a gradient to 3.5 % hex:IPA in 25 min and a column-cleaning step with 10 % IPA in hexane. We scanned for the following masses m/z = 1302.3, 1300.3, 1298.3, 1296.3, 1292.3, 744.0, 1050.0, 1036.0, 1022.0, 1020.0 and 1018.0. Samples were diluted in 500 μl hex:IPA (1.4%) and 5 μl were injected.

## Reporting summary

Further information on research design is available in the Nature Portfolio Reporting Summary linked to this article.

## Data availability

All data are available at PANGAEA https://doi.org/10.1594/PANGAEA. 982881. Supplementary Data 1 contains updated age models to GTS20 for ODP Sites 730, 714, 761, 754, 722, and NGHP-01-01A. Supplementary Data 2 contains trace element data for ODP Sites 730 and 714. Supplementary Data 3 contains foraminifera-bound nitrogen isotope data for ODP Sites 730 and 714. Supplementary Data 4 contains GDGT data for ODP Site 730.

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

## Acknowledgements

This research used samples provided by the International Ocean Discovery Program (IODP). We thank Kaixuan Bu for help analyzing trace elemental compositions, Lois Merritt for her help with trace element sample preparation, Ralf Schiebel for help with foraminiferal identification, and Björn Taphorn for picking foraminifera for foraminifera-bound nitrogen isotope analysis. We thank Paul Falkowski, Fiorella Prada, and Liti Haramaty for their help with scanning electron microscope imaging. We thank Anta-Clarisse Sarr and three anonymous reviewers for reviewing this manuscript. Funding was provided by U.S. Science Support Program Schlanger Fellowship to AVH, a Rutgers Bevier graduate student fellowship to AVH, U.S. National Science Foundation grant OCE-2303513, Swiss National Science Foundation grant P2EZP2_200000 to AA and Max Planck Society funding to A.M.G.

## Author contributions

A.V.H. and A.A. designed the study and wrote the manuscript. A.V.H. measured trace element data in the lab of Y.R. A.A. measured foraminifera-bound nitrogen isotopes and biomarkers in the lab of A.M-G. A.V.H., A.A., Y.R., A.M-G., and D.M.S. contributed to the interpretation and manuscript writing at different stages of the project.

## Funding

## Competing interests

The authors declare no competing interests.
