## [Transparent Peer Review file · Communications Earth & Environment]

Contrasting Evolution of the Arabian Sea and Pacific Ocean Oxygen Minimum Zones during the Miocene

Corresponding Author: Dr Alexandra Auderset

Version 0:

Decision Letter:

Dear Dr Auderset,

Please accept our apologies for the delay in sending a decision on your manuscript. Your manuscript titled "Contrasting Evolution of the Arabian Sea and Pacific Ocean Oxygen Minimum Zones during the Miocene" has now been seen by our reviewers, whose comments appear below. In light of their advice we are delighted to say that we are happy, in principle, to publish a suitably revised version in Communications Earth & Environment.

We therefore invite you to revise your paper one last time to address the remaining concerns of our reviewers. At the same time we ask that you edit your manuscript to comply with our format requirements and to maximise the accessibility and therefore the impact of your work.

EDITORIAL REQUESTS:

****Please take care to match our formatting and policy requirements. We will check revised manuscript and return manuscripts that do not comply. Such requests will lead to delays. ****

SUBMISSION INFORMATION:

OPEN ACCESS:

Communications Earth & Environment is a fully open access journal. Articles are made freely accessible on publication. For further information about article processing charges, open access funding, and advice and support from Nature Portfolio, please visit <https://www.nature.com/commsenv/open-access>

Link Redacted

Best regards,

Alice Drinkwater, PhD
Associate Editor
Communications Earth & Environment
Consulting Editor
Communications Sustainability

REVIEWERS' COMMENTS:

Reviewer #1 (Remarks to the Author):

I have reviewed the revised manuscript and I am happy with the way the authors handled my comments. I think they did make a great job at better explaining the questions they try to address and describing the very diverse tools they use. This is very useful for non-specialist readers as myself. I am also more satisfied with the discussion as is now. I think it highlights better the complexity of the drivers of OMZ evolution especially in this region where it is impacted both by very dynamics local paleogeography and global ocean evolution.

I will be very glad to see this study publish as it is another step forward in deriving a complete picture of the global ocean dynamics during the Miocene, and will for sure motivate additional efforts to better understand driving mechanisms and forcing of Miocene climate change.

Minor comments :

L.276 (tracked change version) "Further deoxygenation following the MCO, between ~14.6 and 12.1 Ma" . I suggest the authors to mentioned it occurred during and after the Middle Miocene Climate Transition as soon as the beginning of the paragraph.

L. 365 (tracked change version) "establishment of the modern southeastern monsoon system". I didn't noticed this when I first reviewed this manuscript but I don't think there is such a thing is as the southeastern monsoon system. The monsoon system in place ocean Indian Ocean/India and most of South Asia is usually referred to as 'South Asian Monsoon system" (by contrast with the East Asian monsoon), or when people refer to the summer upwelling as "Indian summer monsoon" or "South Asian summer monsoon".

L. 367 (tracked change version) "the emergence of the Arabian Peninsula" > emergence of land in the Arabian Peninsula region

L. 367 (tracked change version) "Iranian Plateau" > Iran-Zagros topography

Anta-Clarisse Sarr

Reviewer #2 (Remarks to the Author):

Overall assessment

This is a timely, well-executed study with a thoughtful comparison to the Pacific. The authors present a new multi-proxy Miocene record from the Arabian Sea (planktonic foraminiferal I/Ca, Mn/Ca, FB- $\delta^{15}\text{N}$, and SST) and compare it with the ETNP to assess how global climate and regional circulation/paleoceanography shaped OMZ evolution across the Miocene. The proxy choices are appropriate, and the MCO-MMCT framing is compelling. With a few clarifications—principally in the abstract, figure ordering/brief species-selection rationale, and minor wording/taxonomic fixes. The manuscript is in very good shape. The authors have incorporated prior reviewers' suggestions and strengthened the discussion.

Long, continuous Arabian Sea records combining foraminiferal I/Ca, FB- $\delta^{15}\text{N}$, and Mn/Ca are scarce; this cross-basin comparison with matched proxies is therefore especially valuable and will interest paleo-OMZ, monsoon, and the broader geoscience communities.

I recommend acceptance with minor revisions.

Detailed comments

Abstract (Line 19)—wording

Lovely start, but the clause “oxygen minimum zones (OMZs) have quadrupled in size since the 1950s” mixes metrics. Do you mean OMZ spatial expansion or the global volume of anoxic waters?. The “quadrupled” applies to anoxic water volume since the mid-20th century, not to OMZ extent. Observations indicate OMZs have expanded and shoaled from the mid-20th century to present, but not “quadrupled.” (e.g., Stramma et al., 2008; Schmidtko et al., 2017; Breitburg et al., 2018; Stramma and Schmidtko, 2021).

Please adjust the opening wording to ‘expanded and shoaled’ rather than ‘quadrupled.’

“Tropical ocean oxygen-minimum zones (OMZs) have expanded and shoaled since the mid-20th century, yet their future trajectory remains uncertain.”

Introduction

Line 74: Well written—clear problem definition, well-justified proxies, and a strong connection to the global context.

Figure 1a—latitude/longitude grid

The lat-long gridlines distract from the currents and O₂ field, and the line weights are inconsistent (e.g., 120°E, the equator, and 40°S are thinner; 100°E is missing). Either remove the grid entirely or restyle it as uniform, thin, dashed lines.

Results

As plotted in figure 2, the Arabian Sea I/Ca and FB- $\delta^{15}\text{N}$ trends are hard to read because the Pacific records have much larger amplitudes and set the shared y-axis range. The Pacific curves visually dominate and compress the Arabian Sea variability. Consider moving the current Fig. 3—your new Arabian Sea proxy records (Mn/Ca, I/Ca, FB- $\delta^{15}\text{N}$)—to Fig. 2 so readers see the present-study results first. Then place the current Fig. 2 (Arabian Sea–ETNP comparison) later, aligned with the Discussion section “Contrasting Pacific and AS OMZ responses” (e.g., as Fig. 4).

Line 118: At first mention, please write the full form of the genus—*Dentoglobigerina altispira*—and then abbreviate thereafter as *D. altispira*. Ensure abbreviation across text, figs, and captions (apply similarly to *Trilobatus sacculifer*).

Lines 114-117 Name the species when reporting site-specific I/Ca values; likewise, report $\delta^{15}\text{N}$ with species (e.g., *T. sacculifer*: $\delta^{15}\text{N} = 4.6\text{--}9.9\text{‰}$) so readers can link values to habitat depth and species effects.

Lines 118-120 clear explanation of the low I/Ca values (relatively) of *D. altispira* at 730, which is clearly supported by the $\delta^{15}\text{N}$ in *T. sacculifer*.

Line 120: Species choice (clarification; cf. Reviewer #3)

Could you briefly explain why you selected *D. altispira* and *T. sacculifer* at Site 730 and 714—both mixed-layer taxa—rather than using the surface and subsurface or including a thermocline dweller? For comparison, Hess et al. (2023) in the ETNP paired surface (*D. altispira*) and subsurface (*D. venezuelana*) dwellers to span habitat depth. In this context, using *Neogloboquadrina dutertrei* (a thermocline dweller near the OMZ upper boundary, potentially well suited for I/Ca low-oxygen reconstructions) would help clarify the species selection.

Figure 4. Using SST to infer upwelling/cooling at the Oman–AS margin is reasonable, but please make the attribution explicit. To further link upwelling and strengthened SW monsoon winds at ODP Site 730A, consider adding TOC and benthic $\delta^{13}\text{C}$ (*Cibicides* spp.) from Gupta et al. 2015 (Palaeo3). TOC—peaking around 12–9 Ma—provides an independent export productivity tied to wind-driven upwelling, while $\delta^{13}\text{C}$ *Cibicides* spp. shows a major shift (+1.5 → -0.5‰ at ~13–11 Ma) that indicates old bottom waters and circulation. Overlaying these series on the figure (or in a supplementary panel; if not, discuss TOC in the text) alongside your SST and faunal indices would sharpen the convergence of evidence across the MMCT intervals.

Line 205—wording: Replace ‘felt’ with ‘first recorded’ (or ‘first evident’).

Line 268—Taxonomic correction: Replace *Globigerinoides bulloides* with *Globigerina bulloides*.

Line 366—please remove the redundancy “Miocene” after MCO.

Line 376—phrasing: Use “low I/Ca in planktonic foraminifera.”

Reviewer #3 (Remarks to the Author):

The manuscript by Hess et al., entitled “Contrasting Evolution of the Arabian Sea and Pacific Ocean Oxygen Minimum Zones during the Miocene”, explores the oxygenation changes in Arabian Sea sites ODP 730 (Oman Margin) and ODP 714 (east of the Maldives) during the Miocene and further explains the different evolution of the Arabian Sea and North Pacific OMZs. For that, they mainly used foraminiferal trace elements and bound nitrogen, and this was well supported by GDGT abundance. This is a good approach, and the results look very promising considering the limitations due to recovery and diagenesis in that region in the Miocene sediments. I went through the revisions made by the authors following their comments in the previous version of the manuscript. The authors have done a wonderful job by doing point-by-point revisions, which improved the manuscript quite significantly and may be considered for publishing. I just have one major concern, which I think is also raised by previous reviewers, and needs to be discussed briefly in the main text - The authors used I/Ca ratio from different planktic foraminiferal species, which have contrasting depth habitats (shallow-dwelling for more distal sites) showing different values. Slightly lower values at site 730 may indicate both proximal to OMZ as well as the use of deeper-dwelling species. How do authors account for uncertainty in I/Ca values due to different species and/or due to spatial variation in OMZ conditions?

Also, the font size of the text within the individual panels of Figure 5 looks small. The authors should consider increasing those by at least one unit. May also consider using Ref.1, Ref.2 etc. in place of individual citations.

** Visit Nature Portfolio's author and referees' website at www.nature.com/authors for information about policies, services and author benefits**

Dear Dr. Lavergne and the editorial team,

We are grateful for the thoughtful and constructive reviews of our manuscript, and for the opportunity to revise the manuscript, respond to questions and comments, and to resubmit to *Nature Communications Earth and Environment*.

The following are major changes that we have implemented:

- 1) Expanded our discussion on the local and regional drivers of the Arabian Sea OMZ, including elaborating on the role and oxygenation of the Tethys water, as well as adding the two mechanisms of convective mixing and nutrient supply through Southern Ocean intermediate waters. On the advice of reviewers, Figure S9 has been moved to the main text, as it gives a thorough overview of all the competing processes that are influencing Arabian Sea OMZ dynamics across the study interval, helping us to have a more nuanced discussion about the competing mechanisms driving Arabian Sea oxygenation history.
- 2) Strengthened the introduction with clearer motivation and broader context about the importance of oxygen for biology.
- 3) Added more details on the choice of sites, proxy use, and proxy limitations.

In the text below, we address each of the comments that have been raised. Below, black text is the original comments from reviewers, and green text is the associated response from the authors.

Please also find the accompanying cover letter.

Sincerely,

Drs. Hess and Auderset (co-lead authors) and co-authors

Reviewers' comments:

Reviewer #1 (Remarks to the Author):

In this paper the authors present a new multi-proxy record from the Arabian Sea over the last 20 Ma. They use bound-nitrogen isotopes and elemental ratio (Mn/Ca and I/Ca) measured on planktonic foraminifera, to provide a semi-quantitative assessment of various stage of deoxygenation in the Arabian Sea Oxygen Minimum Zone. Results suggest that the OMZ was better oxygenated during the warmer part of the mid-Miocene (Middle Miocene Climate Optimum) and deoxygenation occurred during the cooling. The new oxygenation record is compared with published datasets from the Equatorial North Pacific OMZ using the same proxy, and further integrated with previous published and new SST records from the tropical Indian Ocean. Authors suggest that the evolution of the Arabian Sea OMZ was controlled both by global climate change and

regional change in paleogeography that have modified the Indian Ocean circulation ; explaining the different timing compared to the Equatorial Pacific OMZ.

Note, that I am not a geochemist and will therefore not comment on the methodology or the validity of proxy interpretation, letting this assessment to another reviewer. The results are to my opinion interesting and worth being published as they will contribute to better understand the Miocene climate dynamics.

The idea that the Arabian Sea OMZ expand after the middle Miocene Climate Optimum is however not new as this is already discussed by Bialik et al (2019) using record from site 722 that is located closely the site 730 where the authors derive the new record.

- We agree that the expansion of the Arabian Sea OMZ following the Middle Miocene Climate Optimum (MCO) has been previously discussed, including by Bialik et al. (2020) using data from Site 722, located near our Site 730. However, our study builds on and significantly extends this work by incorporating multiple geochemical proxies, including the novel oxygenation proxies of I/Ca and foraminifera-bound nitrogen isotopes ($FB-\delta^{15}N$), which are less susceptible to diagenetic alteration than bulk sediment- $\delta^{15}N$ used in previous studies (like Bialik et al., 2019). Additionally, by comparing two sites positioned differently relative to the OMZ core (Site 730 near the core of the OMZ and Site 714 more distal), we provide new spatial context for the evolution of the OMZ. This multiproxy, multi-site approach allows us to better constrain both the timing and mechanisms of deoxygenation across the Arabian Sea during the Miocene.

My main criticism is that for now the discussion is really light and only focus on local dynamics without really discussing how it integrate with global climate changes. The authors however claim several times within the manuscript that part of the signal is related to global climate change. I therefore strongly encourage them to make more efforts in integrating their Arabian Sea record with what is known about the evolution of the Indian ocean basin during the Miocene. There are many records - that are summarized on figure S9 - that cover the same time period in this oceanic basing and I think they should be better integrated to the discussion so that the paper would have an higher impact on the community.

- We appreciate the reviewer's suggestion to better integrate our findings with broader Miocene climate and oceanographic changes across the Indian Ocean basin. Our initial decision to keep the discussion focused on local dynamics was guided by the scope and readership of *Nature Communications*, with an emphasis on identifying the primary drivers of Arabian Sea deoxygenation. However, we agree that placing our results in a wider regional and global context would strengthen the manuscript and broaden its relevance.
- To address this, we expanded the discussion on local and regional effects on Arabian Sea OMZ in Sections "Arabian Sea Hypoxia During the Early Miocene and MCO" and "Regional (De)Oxygenation Mechanisms in the Arabian Sea" and moved Figure S9 into the main text as Figure 5. This figure provides a

synthesized overview of key Miocene records from across the Indian Ocean, including upwelling & productivity, monsoonal dynamics, tectonics & circulation, as well as the reconstructed oxygenation trends that allows us to visually contextualize our findings without overextending the discussion in the text.

Main comments

-l. 219 - The authors mentioned global climate cooling effect here but never describe how global cooling is forcing the regional processes (ie, upwelling, monsoon etc..). I suggest they expend the discussion on this topic as it would make the results discuss here more impactful.

- We added a few paragraphs addressing the links to Miocene climate and the MMCT transition (lines 426-494 and 599-607).

- L231-232 - the authors write that SST remains stable and cooler at distal sites. But this statement do not really correspond to what is shown on figure S8 for individual sites. One striking thing seems to me that site 714 show warming around 14 Ma, while site 761 for example exhibits a cooling at that time. I suggest the authors modify their statement so it reflects the heterogeneity of the open ocean response which is likely to be informative on the nature of climate change recorded at that time. This is divergent response for different sites do not seems super surprising because they all lies in different parts of the equatorial current system. For example: the cooling pattern at site 761 is discussed extensively in Sosdian and Lear (2020) paper and my recollection is that the authors attribute the change in temperature to modification of the circulation in the Indonesian Throughflow. If this theory is true, changing water export through the ITF might trigger change in SST at site 761 but would not necessarily affect site 714 that is further north.

- We changed the language to mention the local heterogeneity and added a short discussion of the cooling step in distal records at ~14 Ma (lines 402-409).

l.274 - the authors write about further restriction the Tethys outflow at 12.1 Ma. But most of previous studies on the region suggest the Tethyan seaway was fully closed by 13.5 Ma (see Straume et al. 2025 for review), which would inhibit strong outflow from Mediterranean sea by then. This age seems to be particularly well constrained and coincide with the increase in GDGTrs on figure S5. The sketch on figure 5b therefore seems to represent an interval <13.5Ma rather than 14.6 to 12.1 Ma.

- We agree, recent reconstructions, including Straume et al. (2025), suggest that the Tethyan Seaway was effectively closed by ~13.5 Ma, restricting any significant Mediterranean outflow thereafter. We now attribute the deoxygenation after 13.5 to increasing upwelling as well as global mechanisms related to MMCT cooling, namely convective mixing due to globally cooler surface temperatures and input of nutrients from the Southern Ocean. We added text to this effect (lines 426-494), and adjusted Figures 4 and 6. The revised text now focuses on other mechanisms for the post-13.5 Ma oxygenation trends.

I.277 - the authors write “Further restriction of Tethys outflow at this time, as indicated by decreasing %GDGTRS, was apparently not enough to counteract this deoxygenation” referring to change at 12 Ma. This is to my opinion not really visible on figure S5 where the rapid changes around 12 Ma seems to mostly reflect the better resolution of the record at that time, which suggest signal might be related to orbital variation that would affect productivity and OMZ in the region but not reflect long term trend. On this plot however, a drastic increase of GDGTs is seen at 12 Ma, which the authors mentioned earlier in the manuscript (line 126). At that place their suggest this might be “related to increased productivity or decreased bottom-water ventilation” but this information never come back in the discussion. Do the authors know of evidence for increase productivity or change in bottom-water ventilation in this region from other studies ?

- We appreciate the reviewer’s observation and agree that the rapid changes in GDGT concentrations around 12 Ma may reflect orbital-scale variability rather than a long-term trend. There are two pronounced changes in %GDGTRS in our record and the first decrease occurs at the same time as changes in Neodymium isotopes, which agrees with the interpretation of restricted Tethys outflow waters. We revised the discussion to clarify this interpretation and adjusted the arrow in Fig. 4 to indicate that the first big change in %GDGTRS occurs at ~13.5 Ma.
 - “At ~13.5 Ma, low-oxygen Tethys outflow constricted due to a drop in sea level, as indicated by a drop in ϵNd at Malta (34) and supported by a dramatic decrease in our %GDGTRS (Fig. 4c).”
 - “The final FB- $\delta^{15}\text{N}$ rise at ~12.1 Ma coincides with a more rapid upwelling increase (Fig. 6c), decreasing oxygenation and leading once again to significant water-column denitrification (Fig. 6c). The combination of decreasing stratification and increased nutrient supply leading to increased productivity may have contributed to deoxygenation.”

The author mostly discuss regional changes associated to Tethys outflow and monsoon as a reason for the change in the $\delta^{15}\text{N}$ in site 730/722 around 12.1 Ma. This peculiar date is however not specific of the Arabian Sea: for example, records from sites 1464 and 1469 along the Australian margin suggest northward displacement of the Southern Hemisphere Westerlies around this time (see Groenveld et al. 2017). This would probably have affected ocean dynamics in the region where intermediate and mode water forms. Those water masses are thought to be supplying tropical Indian Ocean, and this is what is discuss in Auer et al (2023), to explain ecologic changes at site 722. Could part of the signal that the author record be related to change in ventilation from Intermediate Water mass as suggest by Auer et al. ?

- This is possible, and we have added this to the discussion in two parts (lines 489-494 and 599-607) and Figure 6.

-I suggest the authors move the Figure S9 in the main text as it contains all the information that support the discussion. This figure could be better use to provide global (or at least Indian Ocean) context to their finding.

- We agree with the reviewer and moved Fig. S9 to the main text and it is now Figure 5.

Minor comments

I. 43 I think the dot after the reference (11) is a typo and should be removed

- Thank you! Corrected.

I.169 - “and nearly continuous eolian dune deposits from 22 Ma requiring monsoonal winds”. I am not sure the reference to Aeolian deposits is relevant here. Those indeed indicate change in the East Asian monsoon system, which is a system that differs from the South Asian monsoon what drive productivity in Arabian Sea. Betzler et al. (2016) seems to be the only reference that describe the wind driven circulation in the tropical Indian Ocean at that time.

- This is a good point. We’ve gone through the monsoon references to check that they are all relevant for South Asian monsoons. As a result, we removed Guo et al. (2002) from the discussion and from Figure 5 and adjusted the Cliff et al. (2008) information to only include the parts specifically about the South Asian Monsoon, which unfortunately begins at ~17M and doesn’t capture the early Miocene. Betzler et al. (2016) see the wind driven circulation increasing at 12.9Ma, when drift deposits appear in the Maldives, unfortunately not helpful for this part of the manuscript aimed at the pre-MCO hypoxia. However, Beasley et al. (2021) do provide evidence for a proto-South Asian monsoon around 23Ma, consistent with the observed hypoxia.

I. 187 - “may be because Site 714 is deeper, lower in the OMZ” - I suggest the authors specify here what is the depth of those site as they mentioned specific depth for the other sites.

- Site depth (2038 m) has been added

I. 250 - “Beyond monsoonal forcing, the dynamic tectonic setting of the Miocene with shifting”. This sentence seems to suggest that monsoonal forcing and tectonic setting are decorrelated. It reads a bit odd as I. 241-243, just couple of sentences before, the authors mentioned literature that suggest topography plays a role in forcing monsoon evolution. I understand the authors want to separate the monsoon-wind effect from the Tethyan outflow effect but I suggest they rephrase so it’s not confusing.

- Thanks for pointing out this potential confusion. We were trying to say that the dynamic tectonic setting affects monsoons and also ocean gateways. We adjusted the phrasing to “Beyond monsoonal forcing, the dynamic tectonic setting of the Miocene also led to shifting ocean gateways, causing dramatic changes in ocean circulation and likely affected the OMZs on a regional scale.”

I. 257 - The authors mentioned several times the Tethys outflow water without really describing what they refer to. I suggest they introduce a bit more what is the

paleogeographical context in the region and relation to Mediterranean Sea - Indian Ocean at that time.

- We have expanded the explanation of Tethys outflow at its first mention: “Alternatively, Early Miocene Arabian Sea hypoxia could be related to low-oxygen Tethys Ocean outflow. Beginning at ~19.5 Ma, a warm, saline water mass originating in what would become the Mediterranean Sea entered the Arabian Sea via the Red Sea and Persian Gulf. Benthic foraminifera assemblages from both the Indian and Atlantic Oceans suggest that this water mass was low-oxygen.” (lines 169-174)

Figure 3b : I understand that authors want to highlight their own record, but this make the figure very difficult to read. I suggest them to plot all the 3 curves in bold on this figure b so trends are visible.

- Instead of putting all the curves in bold, which we found would also make the figure hard to read, we removed the vertical lines and added small arrows to indicate where the major changes are observed.

Figure” 3c : could the authors explain why they place the step 1) around 14Ma while the decreasing trend is visible since approximatively 15Ma ?

- We take the point of change as between the last data point where there’s no change and the first where there is, since we don’t know what happens between the two points. The line had drifted a bit too far to the left here, which we have corrected.

References :

Betzler et al., 2016 Scientific Reports.
Groenveld et al. 2017, Science Advances
Auer et al 2023 Climate of the Past
Straume et al. 2025 Nature Reviews Earth & Environment
Sosdian and Lear 2020 Paleoclimatology & Paleoceanography

Reviewer #2 (Remarks to the Author):

Haas et al.s’ manuscript titled: Contrasting Evolution of the Arabian Sea and Pacific Ocean Oxygen Minimum Zones during the Miocene aims to track changes in the temporal and spatial changes of oxygen-depleted ranges of the Indo-Pacific through the Miocene. The authors main tools of choice are foraminiferal trace element geochemistry. Specifically, two oxygen-sensitive indexes (Mn/Ca and I/Ca) and one isotope system (foraminifera-bound nitrogen isotopes) as well as biomarkers. All three foraminifera-based proxies are established tracers for oxygen state, with $\delta^{15}\text{N}$ and I/Ca representing the water column and Mn/Ca the bottom water. Samples were obtained

from two locations ODP Site 730 on the Oman Margin and ODP Site 714 on the eastern flank of the Maldives.

Neither site is optimal, both have recovery issues in their lower part which complicate their age model with 730 also has a significant unconformity at its upper part, which could lead to alteration of the sediment post-deposition (more relevant to Mn/Ca). That said, there are no real alternatives for the western Arabian Sea due to IODP refusal to conduct operations in the region for fear of piracy for the entire duration of the program.

- We agree and have to work with what cores are available in this time period and region.

The addition of biomarkers analysis helps mitigate some of the possible diagenetic issues, at least for paleothermometry given only GDGT analysis was report (addition of hopanes and N-alkenes could be beneficial to relate the reported change in oxygen to productivity and stratification patterns).

- Unfortunately, we did not analyze the non-polar fraction for n-alkanes or hopanes and thus cannot get more information on productivity or stratification, but believe that the combination of I/Ca, FB-d15N and GDGT is a strong approach to reconstruct the oxygenation and upwelling dynamics in the Arabian Sea.

The topic is interesting and timely, and would be of interest to a range of readers. The findings here overall support and substantiate the findings from prior works with new tools, offering important validation. Overall, following revisions, this paper would be accepted for publication. See my comments above and line comments below.

I have also reviewed the supplementary material. Figure S9 is excellent, and it should be considered for the main text.

- Thank you, this figure is moved to the main text and is now Figure 5.

Data S2, if in XLSX format, should have the formula in column AB rather than values. Data S4 is insufficient, and the full information for the GDGTs should be included.

- We updated the data files as requested.

I had not seen a repository reference for this data; all tabled that should be additionally uploaded to a repository and linked before publication.

- The data is uploaded to Pangaea and under moratorium, but available upon request during the review process.

Line comments -

Line 42: This is an open debate in the community, but I favor Middle Miocene Climatic Optimum (MMCO) rather than Miocene Climatic Optimum (MCO).

- We prefer to stick with MCO.

Lines 51-59: A clearer statement of the aim of this work / hypothesis to be tested should be included.

- We've added a few sentences in the introduction of the manuscript to state the aim of this work more clearly:
 - "Recent deoxygenation trends exhibit significant spatial variability, with disparate responses in different ODZs (e.g., 16). Consequently, oxygenation pattern from the Miocene ETNP may not be directly applicable to other major ODZs, such as the Arabian Sea. The primary aim of this study is to test whether the Arabian Sea ODZ responded to Miocene climate warming in a similar manner to the ETNP, and to identify the mechanisms driving any observed differences. To achieve this,..."

Lines 60-67: I contemplate the necessity of this paragraph. It feels parts of it could be incorporated into the prior paragraph and streamlined without any loss of information.

- Agreed, we shortened the last two paragraphs and combined the text into one paragraph.

Lines 107-110: Suggest reversing the order, first establish the magnitude or proportion between I/Ca to pO_2 , then describe.

- We updated for each of the proxies in the results section the order of description.

Line 171: Torfstein & Steinberg (2020, SR) reported a marked dropped in eastern Mediterranean mass accumulation rate (for both carbonate and siliciclastic) between ~22 and ~17 Ma which was ascribed to regional reconfiguration, suggesting low productivity. While Zammit et al., (2022, P&P; 2025, JGS) argued specifically for central Mediterranean increase in productivity around 19 Ma, both Taylforth et al. (2014, MPG) and Bialik et al., (2022, Paleo3) both showed initiation of anoxic sediments in the eastern basins did not occur before ~16 Ma and no real change in sedimentation rate. As such, it is not clear if anoxic waters would be exported at these times from the Mediterranean to the east. It may be that the water mass Smart reported might not be strictly Tethyan but represent a source inside the Indian Ocean.

- The evidence the reviewer cites is certainly relevant and interesting, and we have expanded the discussion to include more information on the complexity of the Miocene Tethys conditions. This section now reads "Beginning at ~19.5 Ma, a warm, saline water mass originating in what would become the Mediterranean Sea entered the Arabian Sea via the Red Sea and Persian Gulf (36). Benthic foraminifera assemblages from both the Indian and Atlantic Oceans suggest a shift to low-oxygen conditions beginning ~19.5 Ma, also implying that Tethys outflow waters were low oxygen (1). Organic-rich sapropel-like layers in a section from Malta spanning 19.2-18.6 Ma likewise suggest that the Tethys may have been anoxic at that time (1), though these deposits do not appear until later (~15-

16 Ma) in the eastern Mediterranean (1, 2) and low sedimentation suggests low productivity in that area (1).”

Line 174: Give age range.

- We added “between ~14.6 and 12.1 Ma”

Lines 184-185: In Site U1466 (Bialik et al., 2020, Paleo 3) the decline in Mn/Ca also appears to be around the same time, around the Langhian-Serravallian boundary (their Figure 8). Also, make sure that the curves have been updated to the new age model (Betzler et al., 2018, PEPS).

- We haven’t been able to find a Bialik et al. (2020) in Paleo 3. We have a Bialik et al. (2020) from *Paleoceanography and Paleoclimatology*, but it doesn’t have a Figure 8. We do have Mn/Ca data from Site U1466 from Betzler et al. (2016), which is in our Figure 3 and discussion. If the reviewer can give us an article title or doi, perhaps we can find it.
- Unfortunately, the Betzler data is not available for us to update the age model. We have reached out to the authors but have not heard back.

Line 214: I feel like there is very important context missing here addressed by Auer et al. (2023, *CotP*), which shows there has been in productivity mode in the Arabian Sea ~12.1 Ma, notably the first appearance of diatoms and an increase in the MAR of diatoms.

- We agree and we had not discussed the mechanism of intermediate water mass nutrient supply at all. We’ve added a paragraph addressing this (lines 484-494).

Line 233: Monsoonal winds.

- Changed to “monsoonal wind-driven upwelling”

Lines 500-501: Please specify the nature of the preservation issues – is it dissolution, fragmentation, or overgrowth? This has importance for the evaluation of the pretreatment protocol used.

- Based on the SEM images (Fig. S9), it is dissolution and overgrowth. We note that the overgrowths are key to our ability to use Mn/Ca as a proxy for bottom water oxygenation, and we discuss the possible effects of diagenesis in Supplementary Text 2: Diagenetic influence on trace elements.

Lines 510-511: Please explain why this was not carried out for 714 given that you report in the prior paragraph the preservation state was poor.

- By omitting the reductive cleaning step, overgrowths are preserved and we gain an additional proxy, Mn/Ca, which tells us about bottom water oxygenation. We provide an extensive explanation of diagenetic considerations in Supplementary

Text 2: Diagenetic influence on trace elements. In addition to this, we see no correlation between Mg/Ca and Sr/Ca or Mn/Ca:

Line 515: Why?

- We are not sure what the reviewer is asking here. Reductive cleaning was omitted both because it affects Mg/Ca and I/Ca and because its omission allows us to use Mn/Ca as a proxy for bottom water temperatures.

Line 538: See Gothmann et al., 2015 (GCA), this Mg/Ca must be taken with a grain of salt and is not likely to be constant through the Miocene.

- This is a good point. Importantly, there is a lot of scatter in the Mg/Ca_{sw} dataset, particularly the coral data, which is why some Miocene researchers choose the halite AKA brine inclusions value, as we have done. This is the same methodology as Sosdian et al. (2020) and Yang et al. (2020), whose data is included in our plots.
- To test the effect of a variable Mg/Ca_{sw}, we've drawn a line through the Gothmann et al. (2015) Mg/Ca_{sw} plot (sloped black line), which also seems reasonable on the Tierney et al. (2010) plot, and corrected our data using this variable Mg/Ca_{sw}.

This results in temperatures up to 1.64°C colder than the originals by 8 Ma (compare bold original temperatures to lighter variable-Mg/Ca_{sw} temperatures), which makes the cooling we observe a conservative estimate.

However, we do not interpret the absolute temperatures, only the difference between Arabian Sea and open ocean sites, which would be affected similarly, leaving the difference relatively unaffected. Finally, since we document a 6°C cooling in the Arabian Sea compared to open ocean, applying a Mg/Ca_{sw} correction that shifts the records <1.64°C would not change our conclusions.

- We added an explanation to this effect to the Methods: “If we were to use a variable $Mg/Ca_{\text{seawater Miocene}}$, increasing through the study interval as in (1), Mg/Ca -derived temperatures would be shifted colder for younger samples by $<2^{\circ}\text{C}$, making our estimated cooling conservative. Since this affects both core OMZ sites and non-OMZ sites, the temperature difference between these areas (Fig. 2b SST gradient) would remain relatively unchanged.”

Figure comments -

Figure 1: No comments.

Figure 2: No comments.

Figure 3: b. There is a more detailed Mn/Ca record for Site U1468 in Bialik et al. (2020, Paleo3), Also note that the age models for the Maldives sites have been updated since the 2016 Betzler et al. paper. d. I do wonder why the authors are not comparing their $\delta^{15}\text{N}$ to those in Bialik et al. (2020, P&P) and Ling et al. (2021, Paleo3). While the values would be offsetted, they would be complementary to the records shown here as they trace the same locations but at different water depths.

- Bialik et al. (2020) note that Mn/Ca in non-calcareous samples are enriched relative to calcareous samples, and to avoid lithology-based variability they highlight the calcareous samples, which is what we have plotted.
- We agree that it would be best to update the Betzler age model and plot the Bialik and Ling nitrogen isotope data. Unfortunately, the Betzler data is not available for us to update the age model. We have reached out to the authors but have not received it. Ling et al. (2021) do not provide age model data and their nitrogen isotope data is available only by depth so we are unable to plot it. We do show the Bialik et al. (2020) bulk sediment nitrogen isotope data in Fig. S4c.

Figure 4b. Caution must be applied here with the distal sites due to the inclusion of Site 754. Analysis of adjacent Site 752 (Christensen et al., 2021, GRL; Lyu et al., 2023, P&P) suggests this area was affected by Pacific/Southern Ocean sourced water masses as of the mid to late Miocene.

- We appreciate the reviewer’s note regarding the potential influence of Pacific/Southern Ocean-sourced water masses at Site 754 and acknowledge that this influence may affect the interpretation of SST trends at Site 754. However, due to the limited availability of open-ocean SST records for this time interval, we believe that including Site 754 remains valuable for capturing broader regional patterns. Moreover, the SSTs at Site 754 fall within a similar range to those at Site 761 (Mg/Ca), suggesting that any Southern Ocean influence on surface temperatures was likely minor during the interval of interest.

Figure 4c. I’m not sure why both axes are flipped if the authors wish to show a decline, wouldn’t a line going down and using the proper orientation make more sense?

- We flipped the axes in Fig. 4e in line with the reviewer's suggestion.

General figure comments: the authors refer repeatedly to varying oxygen state based on the threshold of the different proxies. Some sort of illustration may help with keeping track of this. With time in the X axis, pO₂ in the Y axis, and rectangles for the range in each duration.

- Thank you for the suggestion. We have added bars to the top of figure 4 showing intervals of hypoxia and denitrification.

Reviewer #3 (Remarks to the Author):

Hess and co-authors have compared the changes in Arabian Sea oxygen deficient zone with that in the North Pacific, during the Miocene. Given the contrasting views regarding the fate of dissolved oxygen under future warming scenario, such studies are important. The proxies used to reconstruct the dissolved oxygen concentration are robust and used frequently. Having said that, the manuscript has several issues listed below.

- The sites chosen by the authors for OMZ reconstruction are not within the core OMZ/ODZ. Both, the Site 730 off Oman as well as Site 714 in the central equatorial Indian Ocean are outside the intense OMZ. In such a scenario, how do the authors claim to reconstruct the OMZ intensity?
 - Though the secondary nitrate maximum (e.g., Rixen et al., 2014) is indeed further eastward, detached from the coast of Oman, the oxygen concentrations do not show that same pattern and are quite low at Site 730. This can be seen in Figures 1a (minimum oxygen concentration at any depth) and S1b (oxygen concentration at 400m). Below is a version of Figure 1a with 1 $\mu\text{mol/kg}$ contours (dark blue dots are data points, showing that the map is well constrained, including along Oman). Since we are discussing the OMZ, which is defined based on oxygen concentrations as defined in the first paragraph of the introduction, this site is in the core of the OMZ. As shown in the map below, oxygen concentrations at this site are below 5 $\mu\text{mol/kg}$, the threshold for denitrification, making it also in the oxygen deficient zone, defined based on the denitrification, and the high nitrate nitrogen-isotope values are felt in this area (Fig. S1d). Site 730 also has the added benefit of being in the modern upwelling area (Fig. S1a), making it ideal for tracking upwelling with SST, as we do. In contrast, Site 714 is more distal to the OMZ, with minimum oxygen concentrations of $\sim 23 \mu\text{mol/kg}$ (map below) and depleted subsurface nitrate $\delta^{15}\text{N}$ values (Fig. S1d).

- The I/Ca is almost same throughout the studied interval (Figure 2), and same is the case with FB- $d^{15}N$, except a few data points, unlike ETNP. The limited data points in FB- $d^{15}N$, hamper a proper understanding of the variation in OMZ intensity prior and post MCO.

- I/Ca is indeed low ($<2 \text{ umol/mol}$) throughout in relation to the ETNP, which leads to one of the most interesting findings of this manuscript, that the two OMZs behaved differently during the Miocene. We discuss the possibility of diagenesis and provide additional trace elemental data showing that this signal is real in the supplement It was generated in the same lab as the ETNP data, so their contrasting signals can be compared. Though the Arabian Sea values are relatively low, they do have some character, shown in Figure. 3c, which indicates decreasing oxygenation following the MCO. That we see the same pattern in both sites, first in the more proximal site then in the more distal, highlights that the trend is real and is happening regionally, despite its small amplitude.
- The $d^{15}N$ data from Site 730 are limited by the samples in which there were sufficient foraminifera for analysis. We did sample above and below these data but did not find sufficient foraminifera. Fortunately, they do cover the shift from the MCO to the MMCT and tell an interesting story about the start-stop-start of denitrification across the 4 My record.

- What about the Mn/Ca data at Site 730? Even the Site 722 data does not cover the intended interval, making it hard to make out the change prior to MCO.

- Mn/Ca data at Site 730 is not available because those samples were reductively cleaned, removing the material used in this proxy. Samples from Site 714 were analyzed later in the project, when we had eliminated the reductive cleaning step,

which allows us to use the Mn/Ca data. We agree that Site 722 does not cover the optimal part of the record for MCO changes, but this is not our data and is included for completeness with the existing literature.

- $\delta^{13}\text{C}$ at Site 714 was although low during MCO, compared to a couple of data points prior to MCO, it was much lower post MCO and even prior to MCO. The temperature was completely different prior to and post MCO. How do you explain this contrasting response? Is this proxy responding to something different than temperature induced oxygenation, as suggested by the authors?
 - We certainly do not think that Miocene oxygenation changes were driven by the effect of temperature on oxygen solubility, as is happening in the modern. Similarly, though we use SST to reconstruct upwelling, we do not invoke temperature as the driver for oxygenation. The temperature record is quite consistent with the oxygenation history, in the following context: pre-MCO low-oxygen, warm Tethys outflow made the Arabian Sea hypoxic, then shutoff of those waters and increasing upwelling of colder deeper waters drove down oxygen concentrations and cooled the Arabian Sea, as laid out in the Discussion and shown in Figure 5. If the reviewer could point us to a part of the text where this is confusing, we would appreciate the opportunity to clarify.
- Authors use a different species at Site 730 and 714. Given the difference in the depth habitats of the species, the isotopic and elemental signatures in the tests of the species are bound to differ. How do the authors reconcile it?
 - This is a good thing to consider. Though *T. sacculifer* and *D. altispira* are both thought of as mixed-layer species, the latter likely lived deeper (Zou et al., 2021). For $\delta^{13}\text{C}$, the species difference may explain why values are slightly lower for Site 730 (*altispira*, deeper and closer to the OMZ than *T. sacculifer*, see Hess et al., 2025) compared to 714. However, we do not compare the values at the two sites, only the timing of onset of change, when $\delta^{13}\text{C}$ values begin to fall. We added “The offset between sites may be due to the different species used at each site; *D. altispira* measured at Site 730 calcifies in the deep mixed layer, closer to the OMZ and therefore possibly prone to record lower $\delta^{13}\text{C}$ values than *T. sacculifer*, which were measured at Site 730. $\delta^{13}\text{C}$ values below $\sim 2 \mu\text{mol/mol}$ indicate oxygen concentrations $< 90 \mu\text{mol/kg}$ throughout the study interval, based on the species non-specific calibration of Hess et al. (2025).” For Mg/Ca, we used the multispecies equation of Anand et al. (2003), which is widely used and appropriate for studies in which multiple species are used.
 - For $\delta^{15}\text{N}$ this is not a concern as for both core sites we measured *T. sacculifer*. However, to make sure that this signal is also observed in other species, we’ve measured *D. altispira*, *P. mayeri*, *G. mendardii* and observed a similar trend with a species-specific offset which is to be expected for deeper-dwelling foraminifera (see Ren et al., 2012, Limnogeology)
- Except one data point, the timing of shift in Mn/Ca at Site 722 is the same as that at Site 714.

- The Site 722 Mn/Ca data was discussed in detail by Bialik et al. (2020).

• I just wonder why did the authors chose to avoid aligning the cross-section AA' through Site 730? Is it because the dissolved oxygen concentration at this site is much higher than the chosen orientation of the cross-section?

- We chose this line because it is a simple, straight line that goes all the way from the OMZ to the distal Indian Ocean. The location of Site 730 has very low oxygen concentrations, and choosing a more complex cross section does not alter this figure or the conclusions of the paper. Here is a version where the cross section line goes through Sites 730 and 714.

- And the version from Figure 1b for comparison

• The journal being interdisciplinary in nature, it is better to add a few lines in the beginning of ‘Introduction’, about the importance of oxygen for life and the factors controlling its availability in the oceans. Also, add a few sentences about the implications of OMZ/ODZ.

- We start the manuscript now with: “Dissolved oxygen is essential for sustaining marine life and plays an important role in regulating oceanic biogeochemical cycles. However, over...

• Introduction, Paragraphs 2 and 3 seem to be disconnected with a sudden shift from ETNP to Arabian Sea. Please add a few connecting sentences for continuity.

- Thank you for pointing this out. We have added “Recent deoxygenation varies spatially, with disparate responses in different ODZs (e.g., Stramma et al., 2010), so Miocene ETNP oxygenation trends are not necessarily applicable to the other major ODZ in the Arabian Sea.”
- The temperature changes are not mentioned at all under the results section.
 - As the focus of this manuscript is on oxygenation changes, we only discuss those in the results section. Temperature changes are only introduced to help understand the oxygenation changes and are thus presented later, so that there is context in which to understand them. If the journal requires that all results be presented first in the results section, we are happy to do this, though it would make the story not flow quite as well.
 - Assuming that upwelling is mainly responsible for the formation of ODZ in the Arabian Sea is wrong. In fact, it is a combination of several factors including the winter convective mixing, deep water circulation, freshwater influx. If the ODZ was upwelling driven, the most intense ODZ should be in the western Arabian Sea. However, the most intense ODZ is in the northeastern Arabian Sea.

We agree and have added a paragraph about circulation changes and convective mixing: “Convective mixing, driven by surface cooling and increased water density, is another mechanism that can influence oxygenation in the upper ocean of the Arabian Sea. Nowadays, this process is most active during winter monsoon seasons, when strong winds enhance evaporative cooling and promote vertical mixing (1, 2). Similar dynamics have been inferred for the Last Glacial Maximum (LGM), where intensified winter monsoonal winds led to substantial surface cooling and deeper convective overturning, entraining nutrient-rich waters into the surface layer and potentially enhancing productivity (1). The global cooling and temperature reorganization during the MMCT could have led to intensified convective mixing and nutrient supply to the surface, promoting ODZ growth. However, due to the lack of seasonal proxies, this mechanism remains speculative.”

Minor Comments

Line 35, ‘low dissolved oxygen levels’

Line 39, ‘on which we’

Line 43, delete ‘.’ after (11)

Line 53, ‘concentrations, namely’

Line 55, tracks

Line 111, please write the full form of the genus, when beginning a sentence with it.

Line 123, ‘concentration at ODP Site’

- Thank you, this and above minor comments have been incorporated.

Line 171, without data, it is a mere speculation

- True, and this sentence is meant to acknowledge that. We later present data for changes in Tethys outflow.

Line 223, it is too simple explanation for the formation of Arabian Sea ODZ, ignoring factors like convective mixing, restricted circulation and others.

- We acknowledge the mechanism of convective mixing as potentially important, however our interpretation focuses on the processes for which we have direct proxy evidence, like regional upwelling and intermediate water circulation. Nevertheless, we have added a few sentences pointing out that convective mixing could also be a driving factor of the Arabian Sea OMZ during the Miocene, but this mechanism needs to be tested with further proxy data (copied above the “minor comments” section herein).

Line 227-230, The upwelling affects the temperature only during a limited season. The species used for Mg/Ca analysis thrive throughout the year thus incorporating annual SST signatures. How do you assume that the upwelling induced seasonal temperature shift will result in such a large shift in annual average temperature?

- We acknowledge that the Mg/Ca-derived SSTs from surface-dwelling planktonic foraminifera reflect annual mean temperatures, while monsoon-driven upwelling is a seasonal phenomenon. However, modern observations show that upwelling along the Somali-Oman margin induces a substantial seasonal SST drop of up to $\sim 6^{\circ}\text{C}$, which can significantly influence the annual mean SST, especially if the upwelling season is intense and prolonged. The two-step $\sim 6^{\circ}\text{C}$ SST cooling observed at Sites 730 and 722 during the Miocene likely reflects a combination of enhanced seasonal cooling and potential shifts in the annual thermal structure of the upper ocean due to more persistent or intensified upwelling. While we cannot resolve seasonal variability directly from the Mg/Ca records, the spatial SST gradients with cooling in the western Arabian Sea and stable temperatures in the east, lipid biomarker-derived SSTs, alongside increased productivity proxies and faunal shifts, support a scenario of regionally intensified upwelling.

Dear Dr. Drinkwater & Dr. Lavergne,

Thank you for your guidance and for forwarding the reviewers' comments on our manuscript titled "Contrasting Evolution of the Arabian Sea and Pacific Ocean Oxygen Minimum Zones during the Miocene."

We appreciate the constructive feedback and have carefully addressed all points raised by the reviewers. Responses to reviewers are in **green**.

In summary, we:

- Revised the abstract and removed abbreviations.
- Changed figure order as suggested by Reviewer 2.
- Clarified species selection rationale and emphasized trends over absolute values in I/Ca interpretations. For this, we also added supplementary trace element data (see supplementary figure 3).
- Incorporated additional discussion on upwelling indicators.
- Made minor wording and taxonomic corrections throughout.
- Adjusted formatting to comply with journal requirements.

We believe these changes have strengthened the manuscript and improved accessibility for a broad readership. Please find the revised manuscript and completed editorial request table uploaded via the submission system.

Thank you again for your support throughout this process. Please let us know if any further adjustments are needed.

Best regards,

Anya Hess & Alexandra Auderset (on behalf of all co-authors)

REVIEWERS' COMMENTS:

Reviewer #1 (Remarks to the Author):

I have reviewed the revised manuscript and I am happy with the way the authors handled my comments. I think they did make a great job at better explaining the questions they try to address and describing the very diverse tools they use. This is very useful for non-specialist readers as myself. I am also more satisfied with the discussion as is now. I think it highlights better the complexity of the drivers of OMZ evolution especially in this region where it is impacted both by very dynamics local paleogeography and global ocean evolution. I will be very glad to see this study publish as it is another step forward in deriving a complete picture of the global ocean dynamics during the Miocene, and will for sure motivate additional efforts to better understand driving mechanisms and forcing of Miocene climate change.

Minor comments :

I.276 (tracked change version) "Further deoxygenation following the MCO, between ~14.6 and 12.1 Ma" . I suggest the authors to mentioned it occurred during and after the Middle Miocene Climate Transition as soon as the beginning of the paragraph.

- Ok, we did this.

L. 365 (tracked change version) “establishment of the modern southeastern monsoon system”. I didn’t noticed this when I first reviewed this manuscript but I don’t think there is such a thing is as the southeastern monsoon system. The monsoon system in place ocean Indian Ocean/India and most of South Asia is usually referred to as ‘South Asian Monsoon system” (by contrast with the East Asian monsoon), or when people refer to the summer upwelling as “Indian summer monsoon” or “South Asian summer monsoon”.

- Thank you for catching this, we corrected this.

L. 367 (tracked change version) “the emergence of the Arabian Peninsula” > emergence of land in the Arabian Peninsula region

- done

L. 367 (tracked change version) “Iranian Plateau” > Iran-Zagros topography

- done

Anta-Clarisse Sarr

Reviewer #2 (Remarks to the Author):

Overall assessment

This is a timely, well-executed study with a thoughtful comparison to the Pacific. The authors present a new multi-proxy Miocene record from the Arabian Sea (planktonic foraminiferal I/Ca, Mn/Ca, FB- $\delta^{15}\text{N}$, and SST) and compare it with the ETNP to assess how global climate and regional circulation/paleoceanography shaped OMZ evolution across the Miocene. The proxy choices are appropriate, and the MCO–MMCT framing is compelling. With a few clarifications—principally in the abstract, figure ordering/brief species-selection rationale, and minor wording/taxonomic fixes. The manuscript is in very good shape. The authors have incorporated prior reviewers’ suggestions and strengthened the discussion.

Long, continuous Arabian Sea records combining foraminiferal I/Ca, FB- $\delta^{15}\text{N}$, and Mn/Ca are scarce; this cross-basin comparison with matched proxies is therefore especially valuable and will interest paleo-OMZ, monsoon, and the broader geoscience communities.

I recommend acceptance with minor revisions.

- It sounds like Reviewer 2 is new, apologies if we’ve misunderstood. We thank the reviewer for stepping in and for their kind words and helpful comments.

Detailed comments

Abstract (Line 19)—wording

Lovely start, but the clause “oxygen minimum zones (OMZs) have quadrupled in size since the 1950s” mixes metrics. Do you mean OMZ spatial expansion or the global volume of anoxic waters?. The “quadrupled” applies to anoxic water volume since the mid-20th century, not to OMZ extent. Observations indicate OMZs have expanded and shoaled from the mid-20th

century to present, but not “quadrupled.” (e.g., Stramma et al., 2008; Schmidtko et al., 2017; Breitburg et al., 2018; Stramma and Schmidtko, 2021).

Please adjust the opening wording to ‘expanded and shoaled’ rather than ‘quadrupled.’
“Tropical ocean oxygen-minimum zones (OMZs) have expanded and shoaled since the mid-20th century, yet their future trajectory remains uncertain.”

- Thank you, we have changed it to “Ocean oxygen minimum zones (OMZs) have expanded since the mid-20th century, yet their future remains uncertain.”

Introduction

Line 74: Well written—clear problem definition, well-justified proxies, and a strong connection to the global context.

Figure 1a—latitude/longitude grid

The lat-long gridlines distract from the currents and O₂ field, and the line weights are inconsistent (e.g., 120°E, the equator, and 40°S are thinner; 100°E is missing). Either remove the grid entirely or restyle it as uniform, thin, dashed lines.

- This is something strange that happens when it’s saved as an image to insert into the Word document. We hope the layout folks will be able to keep the figure at a sufficient resolution to show the lines consistently, and will pay attention to how the grid lines look after layout. Here is a screenshot of the grid lines zoomed in, before they’re downsampled:

Results

As plotted in figure 2, the Arabian Sea I/Ca and FB- $\delta^{15}\text{N}$ trends are hard to read because the Pacific records have much larger amplitudes and set the shared y-axis range. The Pacific curves visually dominate and compress the Arabian Sea variability. Consider moving the current Fig. 3—your new Arabian Sea proxy records (Mn/Ca, I/Ca, FB- $\delta^{15}\text{N}$)—to Fig. 2 so readers see the present-study results first. Then place the current Fig. 2 (Arabian Sea–ETNP comparison) later, aligned with the Discussion section “Contrasting Pacific and AS OMZ responses” (e.g., as Fig. 4).

- This is a good idea, we’ve moved it to Figure 4. Thank you.

Line 118: At first mention, please write the full form of the genus—*Dentoglobigerina altispira*—and then abbreviate thereafter as *D. altispira*. Ensure abbreviation across text, figs, and captions (apply similarly to *Trilobatus sacculifer*).

- Done, thank you.

Lines 114-117 Name the species when reporting site-specific I/Ca values; likewise, report $\delta^{15}\text{N}$ with species (e.g., *T. sacculifer* $\delta^{15}\text{N} = 4.6\text{--}9.9\text{‰}$) so readers can link values to habitat depth and species effects.

- This is a good idea, but would mean discussing the species earlier, which would shift the emphasis. It is all discussed in the same paragraph, and species are also included in the figure captions, so we feel this information is already easy to find.

Lines 118-120 clear explanation of the low I/Ca values (relatively) of *D. altispira* at 730, which is clearly supported by the $\delta^{15}\text{N}$ in *T. sacculifer*.

Line 120: Species choice (clarification; cf. Reviewer #3)

Could you briefly explain why you selected *D. altispira* and *T. sacculifer* at Site 730 and 714—both mixed-layer taxa—rather than using the surface and subsurface or including a thermocline dweller? For comparison, Hess et al. (2023) in the ETNP paired surface (*D. altispira*) and subsurface (*D. venezuelana*) dwellers to span habitat depth. In this context, using *Neogloboquadrina dutertrei* (a thermocline dweller near the OMZ upper boundary, potentially well suited for I/Ca low-oxygen reconstructions) would help clarify the species selection.

- We started with Site 730 *D. altispira* to be consistent with the Site 845 Hess et al. (2023) data but switched to *T. sacculifer* at Site 714 because, being an extant species, the calibrations for Mg/Ca-derived temperature are more precise.
- We actually did measure I/Ca in *D. venezuelana* at Site 714, and the values are very similar to those of *T. sacculifer*. Hess et al. (2025) showed that in oxygen minimum zones, where I/Ca in surface species is very low (e.g., $<2\text{ }\mu\text{mol/mol}$), the values in both surface and subsurface species collapse to low values, so this is not surprising. That the Mg/Ca for *D. venezuelana* shows colder temperatures than the *T. sacculifer* data supports their relative depth habitats and is encouraging about the trace elemental data being primary. The *D. venezuelana* data doesn't change our interpretation at all and we had originally left it out for simplicity, but we are happy to include it; perhaps it will be useful for future researchers.
- We've now included the *D. venezuelana* data in supplementary figure S3 and added a sentence to the results: "I/Ca values from shallow-subsurface species *Dentoglobigerina venezuelana* from Site 714 are similar to those from *T. sacculifer* (Supplementary Figure 3a), which is expected given the overall low I/Ca values. With different mixed layer species at the two sites, we caution against overinterpreting differences in absolute I/Ca values between sites and instead focus on trends through time."
- Here is the I/Ca and Mg/Ca data from all species, from Supplementary Figure 3:

Fig. S3. Trace elemental data from ODP Sites 714 and 730 in surface dwellers *T. sacculifer* and *Dentoglobigerina altispira* and subsurface dweller *Dentoglobigerina venezuelana*. (a) I/Ca. (b) Mn/Ca. (c) Mg/Ca. (d) Fe/Ca. (e) Sr/Ca. Error bars indicate 1 sigma standard deviation.

Figure 4. Using SST to infer upwelling/cooling at the Oman–AS margin is reasonable, but please make the attribution explicit. To further link upwelling and strengthened SW monsoon winds at ODP Site 730A, consider adding TOC and benthic $\delta^{13}\text{C}$ (*Cibicides* spp.) from Gupta et al. 2015 (Palaeo3). TOC—peaking around 12–9 Ma—provides an independent export productivity tied to wind-driven upwelling, while $\delta^{13}\text{C}$ *Cibicides* spp. shows a major shift (+1.5 → -0.5‰ at ~13–11 Ma) that indicates old bottom waters and circulation. Overlaying these series on the figure (or in a supplementary panel; if not, discuss TOC in the text) alongside your SST and faunal indices would sharpen the convergence of evidence across the MMCT intervals.

- We agree with the reviewer that adding the existing TOC and stable isotope record would strengthen our interpretation. We initially considered adding the TOC and %*G.bulloides* data from 730 to Figure 4. Unfortunately, the data is not readily available. Instead, we added a sentence about the TOC “This trend is also seen in the total organic carbon content in Site 730, peaking around 12-9 Ma.” and “...as well as a benthic $\delta^{13}\text{C}$ shift to lighter values that indicates the influence of old (upwelled) bottom waters at Site 730 “referring to the Gupta paper.
- Another way to highlight the convergence of evidence is Figure 5, which provides a summary of published records and their evidence for upwelling (Fig. 5b), where we included the Gupta paper as well.

Line 205—wording: Replace ‘felt’ with ‘first recorded’ (or ‘first evident’).

- Replaced with “first recorded”

Line 268—Taxonomic correction: Replace *Globigerinoides bulloides* with *Globigerina bulloides*.

- Thank you! Corrected

Line 366—please remove the redundancy “Miocene” after MCO.

- Corrected to “The Miocene provides”

Line 376—phrasing: Use “low I/Ca in planktonic foraminifera.”

- ok

Reviewer #3 (Remarks to the Author):

The manuscript by Hess et al., entitled “Contrasting Evolution of the Arabian Sea and Pacific Ocean Oxygen Minimum Zones during the Miocene”, explores the oxygenation changes in Arabian Sea sites ODP 730 (Oman Margin) and ODP 714 (east of the Maldives) during the Miocene and further explains the different evolution of the Arabian Sea and North Pacific OMZs. For that, they mainly used foraminiferal trace elements and bound nitrogen, and this was well supported by GDGT abundance. This is a good approach, and the results look very promising considering the limitations due to recovery and diagenesis in that region in the Miocene sediments. I went through the revisions made by the authors following their comments in the previous version of the manuscript. The authors have done a wonderful job by doing point-by-point revisions, which improved the manuscript quite significantly and may be considered for publishing. I just have one major concern, which I think is also raised by previous reviewers, and needs to be discussed briefly in the main text -

The authors used I/Ca ratio from different planktic foraminiferal species, which have contrasting depth habitats (shallow-dwelling for more distal sites) showing different values. Slightly lower values at site 730 may indicate both proximal to OMZ as well as the use of deeper-dwelling species. How do authors account for uncertainty in I/Ca values due to different species and/or due to spatial variation in OMZ conditions?

- This is an important point. We have been careful not to interpret the absolute values of I/Ca between sites because species vary between sites. Instead, we use trends in I/Ca at each site, in conjunction with trends in other oxygenation proxies (Mn/Ca, FB- $\delta^{15}\text{N}$). To be more explicit, we added “With different mixed layer species at the two sites, we caution against overinterpreting differences in absolute I/Ca values between sites and instead focus on trends through time.”

Also, the font size of the text within the individual panels of Figure 5 looks small. The authors should consider increasing those by at least one unit. May also consider using Ref.1, Ref.2 etc. in place of individual citations.

- Thank you for the helpful suggestion. We have increased the font size within the panels of Figure 5 to improve readability, as recommended.
- Regarding the citation format, we appreciate the reviewer’s input. However, we have opted to retain the full reference format (e.g., *Bialik et al., 2020*) rather than using abbreviated forms like *Ref. 1*, *Ref. 2*. This decision is based on the potential for the figure to be used independently of the paper in future presentations or educational contexts, where full citations provide clearer attribution and facilitate easier reference to the original sources.